# Biomarker Proxy Records of Arctic Climate Change During the Mid-Pleistocene Transition from Lake El'gygytgyn (Far East Russia)

Kurt R. Lindberg[1,2], William C. Daniels[1], Isla S. Castañeda[1], Julie Brigham-Grette[1]

[1]University of Massachusetts Amherst, Amherst, MA, 01003, U.S.A.
[2]Now at: University at Buffalo, Buffalo, NY, 14260, U.S.A.

*Correspondence to*: Kurt R. Lindberg (kurtlind@buffalo.edu)

**Abstract.** The Mid-Pleistocene Transition (MPT) is a widely recognized global climate shift occurring between approximately 1,250 to 700 ka. At this time, Earth's climate underwent a major transition from dominant 40 kyr glacial-interglacial cycles to quasi-100 kyr cycles. The cause of the MPT remains a puzzling aspect of Pleistocene climate. Presently, there are few, if any, continuous MPT records from the Arctic, yet understanding the role and response of the high latitudes to the MPT is required to better evaluate the causes of this climatic shift. Here, we present new continental biomarker records of temperature and vegetation spanning 1,142 to 752 ka from Lake El'gygytgyn (Far East Russia). We reconstruct warm-season temperature variations across the MPT based on branched glycerol dialkyl glycerol tetraethers (brGDGTs) using the MBT′$_{5ME}$ proxy. The new Arctic temperature record does not display an overall cooling trend during the MPT but does exhibit strong glacial-interglacial cyclicity. Spectral analysis demonstrates persistent obliquity and precession pacing over the study interval and reveals substantial sub-orbital temperature variations at ~900 ka during the first "skipped" interglacial. Interestingly, Marine Isotope Stage (MIS) 31, which is widely recognized as a particularly warm interglacial, does not exhibit exceptional warmth in the Lake El'gygytgyn brGDGT record. Instead, we find that MIS 29, 27 and 21 were as warm or warmer than MIS 31. In particular, MIS 21 (~870 to 820 ka) stands out as an especially warm and long interglacial in the continental Arctic while MIS 25 is a notably cold interglacial. Throughout the MPT, Lake El'gygytgyn pollen data exhibit a long-term drying trend, with a shift to an increasingly open landscape noted after around 900 ka (Zhao et al., 2018), which is also reflected in our higher plant leaf wax (*n*-alkane) distributions. Although the mechanisms driving the MPT remain a matter of debate, our new climate records from the continental Arctic exhibit some similarities to changes noted around the North Pacific region. Overall, the new organic geochemical data from Lake El'gygytgyn contribute to expanding our knowledge of the high-latitude response to the MPT.

## 1 Introduction

Since the start of the Industrial Revolution, high-latitude temperatures have increased at about twice the global average rate (Serreze et al., 2009; Davy et al., 2018). Likewise, during past interglacial periods, Arctic temperature reconstructions indicate significant warming events (e.g., de Wet et al., 2016). The importance of studying past Arctic temperature variability is widely recognized (Miller et al., 2010; Melles et al., 2012; Brigham-Grette et al., 2013) yet few long and continuous records exist from the continental Arctic. Lake El'gygytgyn, located in Far East Arctic Russia, is a unique site that escaped continental glaciation and preserves a 3.6 myr-long sedimentary record (Nolan and Brigham-Grette, 2007; Melles et al., 2012). Prior work on Lake El'gygytgyn has revealed exceptional changes during the Plio-Pleistocene. During the Pliocene, cool mixed forest and cool conifer forest dominated the landscape (Brigham-Grette et al., 2013), whereas today the tree line lies ~150 km to the southwest and the lake is surrounded by tundra vegetation (Brigham-Grette et al., 2007). In addition to notable changes between Pliocene and Pleistocene climates (Brigham-Grette et al., 2013; Melles et al., 2012),

significant climate variability within the Pleistocene is documented at Lake El'gygytgyn (Melles et al., 2012; Wennrich et al., 2013; Francke et al., 2013), including the presence of numerous exceptionally warm "super interglacial" periods, which pollen spectra suggest were characterized by elevated temperature (4-5 °C warmer) and precipitation (up to 6 times higher) compared to the Holocene (Melles et al., 2012). Prior studies of Lake El'gygytgyn have also noted a signal of the Mid-Pleistocene Transition (MPT) (Melles et al., 2012; Francke et al., 2013; Wennrich et al., 2014), a globally recognized event that occurred from 1.2–0.6 Ma when variations in global ice volume shifted from exhibiting a dominant 41 kyr periodicity to a 100 kyr periodicity (Past Interglacial Working Group of PAGES, 2016).

The cause of the MPT remains a highly debated and puzzling aspect of Pleistocene climate. Between 900 and 650 ka, glacial-interglacial cycles grew longer, more intensified, and asymmetric in association with increasingly large northern hemisphere glacial-stage ice sheets (Maslin and Ridgewell, 2005 and references therein). The MPT cannot be attributed to changes in Earth's orbital parameters (eccentricity, obliquity, precession) and thus the cause is believed to be internal to the global climate system (e.g., Berger and Loutre, 1991; Maslin and Ridgewell, 2005). Numerous, non-exclusive hypotheses have been proposed to explain the MPT. These include gradual removal of regolith from the northern high latitudes allowing for greater vertical growth of the northern ice sheets by increasing basal friction (Roy et al., 2004; Clark and Pollard, 1998; Willeit et al., 2019), an increase in Antarctic ice volume (Pollard and DeConto, 2009; Elderfield et al., 2012; Billups et al., 2018), and significant changes in Atlantic (Poirier and Billups, 2014), Pacific (Martínez-Garcia et al., 2010) or Southern Ocean circulation (Hasenfratz et al., 2019; Rodríguez-Sanz et al., 2012; Pena and Goldstein, 2014). Gradual atmospheric carbon dioxide ($pCO_2$) drawdown and associated climatic cooling are also commonly cited causes of the MPT (Clark et al., 2006; Raymo, 1997; Saltzman and Verbitsky, 1993; Paillard, 1998; Hönisch et al., 2009; Willeit et al., 2019) although empirical evidence for declining $pCO2$ levels across the early and mid-Pleistocene remain equivocal (Da et al., 2019). As a corollary to the $pCO2$ explanation, it has been suggested that the North Pacific and Bering Sea region may have played an important role. For example, the closure of the Bering Strait combined with expanded sea ice cover at 920 ka may have suppressed $CO_2$ ventilation from the subarctic North Pacific to the atmosphere (Kender et al., 2018; Worne et al., 2020). Likewise, Müller et al. (2018) propose that changes in iron fertilization to the northeast Pacific from glaciogenic Alaskan dust and ice rafting helped to drive global climate changes noted during the MPT.

A complete understanding of the Arctic's role in the MPT requires continuous paleotemperature records spanning this interval. In this study, we examine the organic geochemistry of Lake El'gygytgyn sediments from 1,142 to 752 ka to reconstruct continental Arctic temperature and environmental variability during the MPT. Specifically, we use branched glycerol dialkyl tetraethers (brGDGTs; Sinninghe Damsté et al., 2000; Weijers et al., 2007), bacterial membrane lipids, to reconstruct past temperature and the distribution of long-chain *n*-alkanes, biomarkers of terrestrial higher plants, to examine past vegetation shifts (Bush and McInerney, 2013). The MPT has been widely documented at globally distributed marine sites (e.g., Clark et al., 2006) and in continental loess records from Asia and Europe (Heslop et al., 2002; Han et al., 2012).

However, at present, relatively few lacustrine records spanning the MPT have been drilled. These include Lake Malawi (Scholz et al., 2007; Cohen et al., 2007), Lake Baikal (Prokopenko et al., 2006), and Lake Ohrid (Wagner et al., 2014; Just et al., 2019); Lake El'gygytgyn is the only such record from the Arctic (Melles et al., 2012; Brigham-Grette et al., 2013). Investigating the cause(s) of the MPT is beyond the scope of this study. However, our new records provide important information regarding high-latitude continental temperature during this critical climate transition. This study also provides new insights on the strength and duration of past interglacials in the continental Arctic. Understanding past Arctic temperature variability is particularly important for placing the current warming into context and for improving models of future climate.

## 2. Study Location

Lake El'gygytgyn (67.5 °N, 172.1 °E, 492 masl) is located within the Anadyr Highlands of northeast Russia, approximately 150 km south of the Arctic Ocean coast (Fig. 1). It is a crater lake, formed following a meteorite impact at 3.58 Ma (Layer, 2000), that has accumulated a continuous sedimentary record since its formation. The 318 m sedimentary sequence was collected in 2009 (Melles et al., 2012; Brigham-Grette et al., 2013), providing a uniquely long Arctic paleoclimate record including the MPT interval (Melles et al., 2012; Brigham-Grette et al., 2013; Nowaczyk et al., 2013; Zhao et al., 2018). The age model for the Lake El'gygytgyn drill core was previously published by Nowaczyk et al. (2013) and is applied here. This age model is based on iterative tie-point identifications using 1) paleomagnetic reversals, 2) comparison of biogenic silica to the LR04 benthic oxygen isotope stack (Lisiecki and Raymo, 2005), and 3) comparison of total organic carbon (TOC) and magnetic susceptibility to summer insolation (Laskar et al., 2004) and has an age precision of ~500 years relative to the insolation reference curve (Nowaczyk et al., 2013). During our study interval, paleomagnetic reversals/excursions provide age constraints at 0.780 Ma, 0.991 Ma, 1.0142 Ma, 1.0192 Ma, and 1.075 Ma (Haltia and Nowaczyk, 2014; Nowaczyk et al., 2013).

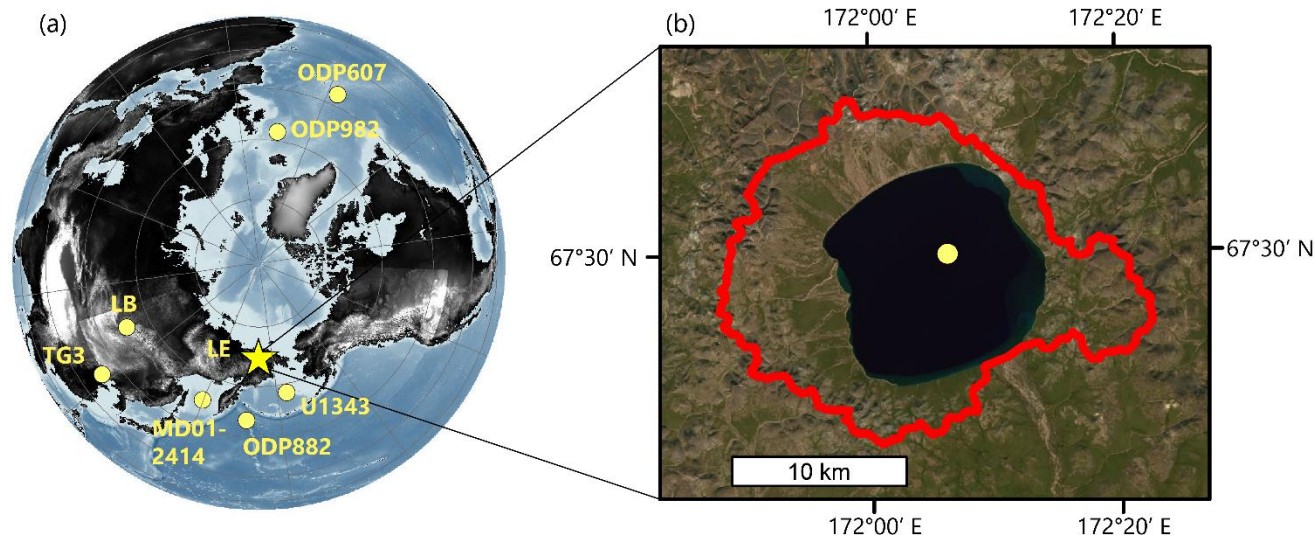

Figure 1: Study location. a) Map showing the location of Lake El'gygytgyn (star) and other key sites discussed in the text (yellow circles) including Ocean Drilling Program (ODP) Sites 882 (Martínez-Garcia et al., 2010), 607 (Sosdian and Rosenthal, 2009), 982 (Lawrence et al., 2009) and International Ocean Discovery Program (IODP) Site U1343 (Kender et al., 2018), MD01-2414 (Lattaud et al., 2018), and Tianjin-G3 loess profile (Zhou et al., 2018). B) Satellite image of Lake El'gygytgyn. The yellow dot represents the coring location of ICDP Site 5011-1 in the central lake basin and the red outline designates the lake's watershed. World imagery sources: Esri, Maxar, GeoEye, Earthstar Geographics, CNES/Airbus DS, USDA, AeroGRID, IGN, and the GIS User Community.

The lake basin morphology and local meteorology are well-characterized by Nolan and Brigham-Grette (2007). The lake is approximately 12 km wide and 175 m deep and resides in an impact crater 18 km in width. There is a single stream outlet, the Enmyvaam River, which drains to the south into the Bering Sea. Approximately 50 small streams, with headwaters located within the impact crater, drain into the lake. Mean annual air temperature is -10.3 °C and summer temperatures (JJA) average 10 °C. In contrast, the lake water never exceeds 4 °C except over shallower regions and in the fringing lagoons which reach 5-6 °C during summer (Nolan and Brigham-Grette, 2007). Over the past 50 years, air temperatures have risen by over 3 °C, driven largely by increasing winter temperatures (Nolan et al., 2013). While positive air temperature anomalies are associated with a strong low-pressure system over the Bering Sea and high pressure over the Beaufort Sea, which advects warm air from the south and east, Nolan et al. (2013) demonstrated that over the observational period, this temperature increase is caused by general warming of the atmosphere, rather than changes in storm tracks. It is somewhat unclear if mean weather patterns at Lake El'gygytgyn are subject to reorganization over longer timescales, such as might be associated with Bering Land Bridge exposure or submergence, or changes in North Pacific Ocean circulation.

## 3. Methods

In this study, we analyzed 127 samples from the Lake El'gygytgyn drill core (ICDP Site 5011-1) spanning the interval from 1,142 ka to 752 ka, corresponding to depths of 46.77 m to 31.06 m in the composite profile. Additionally, we re-analyzed approximately every third sample from the study of de Wet et al. (2016) (41 samples), who also studied Site 5011-1 in the interval from Marine Isotope Stage (MIS) 33 to MIS 31 (1.11 to 1.05 to Ma), using a newer High-Performance Liquid Chromatography (HPLC) method that separates the 5- and 6-methyl brGDGT isomers (Hopmans et al., 2016). We also

include 85 $n$-alkane samples from another study by de Wet (2017) in our analysis. Overall, the sample spacing averages 2.3 kyr for the brGDGTs and 2.7 kyr for the $n$-alkanes, with each 1 cm interval integrating an average 450 years, thereby allowing sufficient resolution to evaluate orbital-scale climate variability.

### 3.1 Biomarker Analyses

Sediment samples were freeze-dried, homogenized, and extracted using a Dionex accelerated solvent extractor (ASE 200)

with a mixture of 9:1 of dichloromethane:methanol (v/v). Known quantities of a synthetic $C_{46}$ GDGT were added to the total lipid extract (TLE) as an internal standard (Huguet et al., 2006). The TLE was then separated via alumina oxide columns using 9:1 hexane:dichloromethane (v/v) to elute the apolar fraction and 1:1 dichloromethane:methanol (v/v) to elute the polar fraction.

### 3.1.1 $n$-alkane analysis

Apolar fractions were analyzed on an Agilent 7890A gas chromatograph-flame ionization detector (GC-FID) to determine concentrations of the $n$-alkanes. Samples were run in splitless mode with an inlet temperature of 250 °C and inlet flow rate of 26.5 mL min$^{-1}$, using hydrogen as the carrier gas. Separation was achieved using a 5% phenyl methyl siloxane column (HP-5; 60 m x 320 μm x 0.25 μm), with a flow rate of 4.6 mL min$^{-1}$. The oven temperature program was as follows: 70 °C for two minutes, increasing at 17 °C min$^{-1}$ to 130 °C, then increasing at 7 °C min$^{-1}$ to 320 °C, and finally holding at 320 °C for

15 minutes. Compound identification was performed by comparison to a standard mixture of $C_{21}$-$C_{40}$ $n$-alkanes injected during each run and by confirming compound identification for a subset of samples via GC-mass spectrometry, following the methods detailed in Keisling et al. (2017). Concentrations of each $n$-alkane were determined using an external squalene calibration curve. Here, we examine leaf wax distributions using the Average Chain Length (ACL; Bray and Evans, 1961; Bush and McInerney, 2013), calculated for the terrestrial ($C_{27}$ to $C_{33}$) $n$-alkanes (Eq. 1):

(1) $\text{ACL} = \sum(C_n * n) / \sum(C_n)$

### 3.1.2 GDGT analysis

The polar fractions were filtered through a 0.45 μm PTFE filter in 99:1 hexane: isopropanol (v/v). Isoprenoid glycerol dialkyl glycerol tetraethers (iGDGTs) and brGDGTs were analyzed using an Agilent 1260 Ultra High-Performance Liquid

Chromatograph (UHPLC) coupled to an Agilent 6120 single quadrupole mass selective detector (MSD) and following the
methods detailed by Hopmans et al. (2016), which separates the 5- and 6-methyl brGDGT isomers. Briefly, separation is
achieved using a silica precolumn with two BEH HILIC columns in series (2.1 x 150 mm, 1.7 μm; Waters®). Samples were
eluted using hexane (solvent A) and hexane:isopropanol (9:1 v/v; solvent B) in the following program: 18% solvent B for 25
minutes, a linear increase to solvent B over 25 minutes, then a linear increase to 100% solvent B for 30 minutes. The column
temperature was 30°C and the flow rate was 0.2 ml min$^{-1}$. Compounds were ionized using atmospheric pressure chemical
ionization and the MSD was run in single ion monitoring (SIM) mode for the GDGT core lipids.

Lake El'gygytgyn sediments are dominated by brGDGTs with only small concentrations of iGDGTs present (Fig. S4)
(Holland et al., 2013; D'Anjou et al., 2013; de Wet et al., 2016; Keisling et al., 2017; Daniels et al., 2021). Thus, we use the
Methylation of Branched Tetraethers index (Eq. 2), considering only the 5-methyl isomers (MBT'$_{5ME}$), to reconstruct past
temperature (De Jonge et al., 2014).

$$(2) \ MBT'_{5ME} = \frac{[Ia + Ib + Ic]}{[Ia + Ib + Ic] + [IIa + IIb + IIc] + [IIIa]}$$

We explored the use of several lacustrine brGDGT calibrations to convert MBT'$_{5ME}$ values to temperature. The first is based
on a suite of lakes in East Africa that span an elevation transect to capture temperature gradients (Russell et al., 2018; Eq. 3).
This calibration is to mean annual air temperature.

$$(3) \ MAAT = -1.21 + 32.42 * MBT'_{5ME} \ \ (RMSE = 2.44 \ °C)$$

More recently, Zhao et al. (2021) developed an in-situ calibration to summer (JJA) water temperature using sediment trap
data from a site in southern Greenland (Eq. 4).

$$(4) \ JJA \ Water \ Temp = -1.82 + 56.06 * MBT'_{5ME} \ (RMSE = 0.58 \ °C)$$

Raberg et al. (2021) explored relationships between compound fractional abundances (FAs) within structural groups based
on methylation number, methylation position, and cyclization number and environmental parameters using a previously
published globally distributed data and adding new sites from the high-latitudes. They recommend using their "Meth"
calibration for general use in lake sediments (Eq. 5):

$$(5) \ MAF \ (°C) = 92.9(\pm15.98) + 63.84(\pm15.58) \times fIb^2_{Meth} - 130.51(\pm30.73) \times fIb_{Meth} - 28.77(\pm5.44) \times$$
$$fIIIa^2_{Meth} - 72.28(\pm17.38) \times fIIb^2_{Meth} - 5.88(\pm1.36) \times fIIc^2_{Meth} + 20.89(\pm7.69) \times fIIIa^2_{Meth} - 40.54(\pm5.89) \times$$
$$fIIIa_{Meth} - 80.47(\pm19.19) \times fIIIb_{Meth} \ (RMSE = 2.14 \ °C)$$

Raberg et al., 2021 also provide a "Full" calibration (Eq.6), provided the highest R$^2$ and lowest RMSE in their data and
suggest this calibration may be appropriate to apply at sites with good conductivity or pH control.

$$(6) \ MAF \ (°C) = -8.06(\pm1.56) + 37.52(\pm2.35) \times fIa_{Full} - 266.83(\pm98.61) \times fIb^2_{Full} + 133.42(\pm19.51) \times fIb_{Full} +$$
$$100.85(\pm9.27) \times fIIa'^2_{Full} + 58.15(\pm10.09) \times fIIIa'^2_{Full} + 12.79(\pm2.89) \times fIIIa_{Full} \ (RMSE = 1.97 \ °C)$$

We also use a mean annual air temperature calibration from Feng et al. (2019) which was developed using lake sediments

and several decades of instrumental climate data from Tiancai Lake, China (Eq. 7):

(7) $MAAT\ (°C) = 14.74 - 34.46 \times f(IIIa) + 27.49 \times f(IIa) - 35.56 \times f(IIb) - 60.36 \times f(1a) - 95.91 \times f(Ib)$

(RMSE = 0.18 °C)

We also apply the global Bayesian calibration for lacustrine brGDGTs, known as BayMBT, using the baymbt_predict()

MATLAB function (Martínez-Sosa et al., 2021). Following the approach of Martínez-Sosa et al. (2021), we estimated the

prior value for our dataset by calculating a mean temperature using the Russell et al. (2018) calibration; we also used 10°C

for the standard deviation.

## 4. Results

### 4.1 *n*-alkanes

Plant leaf waxes (*n*-alkanes) were present in all samples. However, 58 of the 212 samples analyzed (27%) contained an

unresolved complex mixture in the apolar fractions, hampering peak identification without further sample purification. The

*n*-alkanes were not quantified for these samples, resulting in lower sample resolution record for the leaf waxes compared to

the GDGTs. For the 153 samples where *n*-alkanes could be sufficiently resolved, samples contained $C_{20}$ through $C_{33}$ *n*-

alkanes, with the shorter chain lengths not always present in each sample. The total concentration of odd-numbered *n*-

alkanes ranges from 0.6 to 60.8 μg g$^{-1}$, with a mean concentration of 4.5 μg g$^{-1}$. Odd-numbered chain lengths dominate,

particularly the $C_{23}$ and $C_{27}$ homologues (Fig. S1), and the carbon preference index (CPI; Eq. S1) over the $C_{25}$-$C_{33}$

compounds averages 3.6 (±0.6 s.d.). When considering all *n*-alkanes from $C_{17}$ to $C_{33}$, the ACL averages 26.2 (±0.9 s.d.). The

$C_{27}$-$C_{33}$ ACL (Eq. 1) varied between 28.5 and 30.1 (Fig. 2a), with a mean of 29.4. The $C_{27}$-$C_{33}$ ACL exhibits a clear shift at

around 900 ka (Fig. 2, Fig. S2); samples older than 900 ka samples are characterized by lower ACL values whereas samples

younger than 900 ka exhibit higher ACL values (Fig. 2b). This shift in ACL values appears to track changes in the pollen-

derived landscape openness index (Zhao et al., 2018), giving rise to a negative correlation between these metrics (Fig. S3).

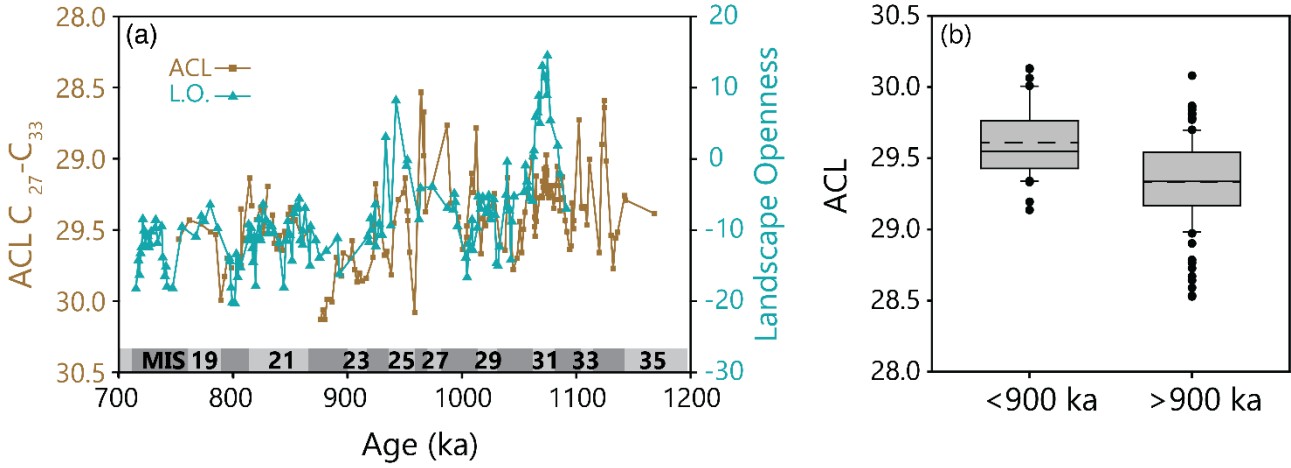

**Figure 2: Comparison of Lake El'gygytgyn *n*-alkane and pollen data. A) The ACL of the $C_{27}$-$C_{33}$ *n*-alkanes plotted along with the Landscape Openness (L.O.) Index derived from pollen spectra (data from Zhao et al. (2018)). B) Boxplots of the ACL data for all samples younger than 900 ka and all samples older than 900 ka. The solid line represents the median value and the dashed line the mean. Individual outliers are plotted.**

## 4.2 GDGTs

iGDGTs are present in all samples analyzed. However, total iGDGT concentrations are low and range from 0.3 ng g$^{-1}$ to 3916 ng g$^{-1}$ with a mean concentration of 69 ng g sed$^{-1}$ (Fig. S4). Nearly all the iGDGTs present are represented by GDGT-0 and GDGT-4 (Daniels et al., 2021), with average abundances of 75% and 15% of the total iGDGTs. The high GDGT0/crenarchaeol ratio (average 68.8), as well as the fact that many samples did not contain all of the iGDGTs required to calculate a TEX$_{86}$ value, precludes the use of the TEX$_{86}$ paleothermometer (Schouten et al., 2002) at Lake El'gygytgyn. This result is consistent with prior investigations of different time intervals of the Lake El'gygytgyn record (D'Anjou et al., 2013; Holland et al., 2013; de Wet et al., 2016; Keisling et al., 2017; Daniels et al., 2021).

Within the entire 174 sample dataset, including the re-analyzed samples from de Wet et al. (2016), total brGDGT concentrations vary from 15 to 4561 ng g sed$^{-1}$ with a mean concentration of 237 ng g sed$^{-1}$ (Fig. S4). Like other Arctic lake sediment records (Zhao et al., 2021; Thomas et al., 2018; Raberg et al., 2021; Peterse et al., 2014) hexamethyl brGDGTs (IIIa, IIIb, IIIc) dominate the brGDGT assemblages (46%), followed by pentamethyl brGDGTs (IIa, IIb, IIc, 30%), then tetramethyl brGDGTs (Ia, Ib, Ic, 23%) (Fig. S4, S5). We observe 6-methyl GDGT isomers in all samples, which comprise an average of 9% of the total brGDGTs present but can be as high as 33% and exhibit substantial variability (Fig. S6). With typically low 6-methyl isomers present, we find very strong correlations in fractional abundances, as well as between MBT′ and MBT′$_{5ME}$ values when comparing between the samples analyzed by de Wet et al. (2016) and then reanalyzed using updated chromatographic methods (Fig. S7). MBT′$_{5ME}$ values range from 0.12 to 0.56 with an average of 0.28. Based on

triplicate analysis of 12 samples, the analytical uncertainty (1σ) of the MBT′$_{5ME}$ is 0.0065. Typically, glacial-interglacial changes in MBT′$_{5ME}$ are on the order of 0.1 to 0.15 units, equivalent to 5-8°C on the Greenland lakes calibration (Zhao et al., 2021), ca. 3-5°C on both the African lakes (Russell et al., 2018) and BayMBT (Martínez-Sosa et al., 2021) calibrations, and are >10°C on the "Full" calibration of Raberg et al. (2021) (Fig. S8). The BayMBT temperature reconstruction of months above freezing is shown in Fig. 3.

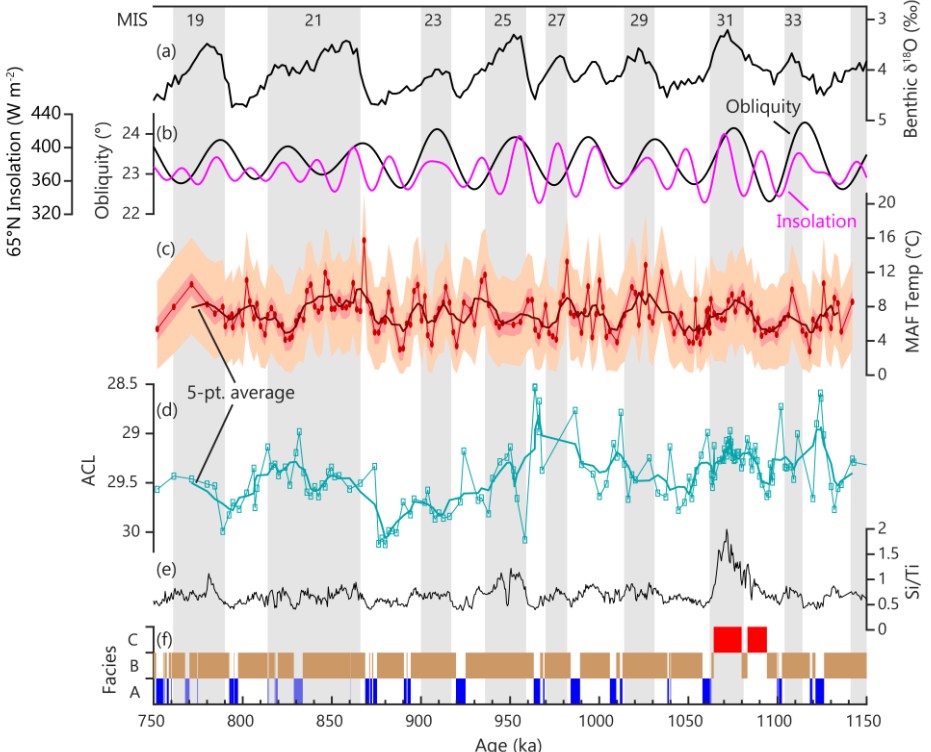

**Figure 3: Lake El'gygytgyn data with Marine Isotope Stages (MIS) notated at the top and with interglacials indicated by the grey shading. A) The global benthic oxygen isotope stack (Lisiecki and Raymo, 2005). B) Astronomical forcing, including obliquity and summer insolation (June 21-Sept 21) at 65 °N latitude (Laskar et al., 2004). C) brGDGT-inferred temperature of the months above freezing (MAF Temp). Dark shading represents the 1σ analytical uncertainty based on triplicate MBT′$_{5ME}$ analyses converted to temperature using BayMBT (1σ =1.35 °C, n=12), while light shading represents the calibration uncertainty (Martínez-Sosa et al., 2021). D) *n*-alkane ACL based on C$_{27}$-C$_{33}$. E) Lake El'gygytgyn silica (Si) to titanium (Ti) ratio (Melles et al., 2012; Wennrich et al., 2013); F) Summary of lithological facies in core 5011-1. Facies A (blue bars) represents glacial intervals. Facies B (brown bars) is cosmopolitan, occurring during both glacials and interglacials. Facies C (red bars) represents superinterglacials.**

## 4.3 Spectral Signatures of Temperature Variability

To characterize the frequency of temperature variability, we analyzed spectral signatures of the MBT′$_{5ME}$ temperature record using PAST3 software (Hammer et al., 2001). Frequency analysis was not performed on the *n*-alkane data because of its lower sample resolution. As noted above, the age model for the El'gygytgyn core tunes TOC and magnetic susceptibility to

240 summer insolation, and indeed, orbital frequencies are observed in several El'gygytgyn proxies (Nowaczyk et al., 2013; Francke et al., 2013). Interestingly, however, the relative strength of precession, obliquity, and eccentricity bands appear to differ between proxies. Here, we use the REDFIT package (Schulz and Mudelsee, 2002) in PAST3 (Hammer et al., 2001), which utilizes the Lomb-Scargle method for spectral analysis (Lomb, 1976; Scargle, 1982; Schulz and Mudelsee, 2002), to evaluate the overall periodogram. We use the PAST3 Short-time Fourier transform package to generate the evolutionary

periodogram. Over our study interval, the REDFIT periodogram shows a small spectral peak at ~82 kyr/cycle, a peak at ~41 kyr/cycles, a dispersed peak in the 23-14 kyr/cycle, as well as a significant peak at ~11 kyr/cycle possibly reflecting half-precession (Fig. 4a). The 40 kyr obliquity signal is strongest in the earlier part of the temperature record, from MIS 32 to MIS 26, but weakens by ~980 ka, when the relative power of the higher-frequency bands increases (Fig. 4b). Toward the later part of the record, beginning at ~900 ka, we observe increased variability at longer wavelengths, potentially reflecting

both 40- and 80-kyr cyclicity (e.g., obliquity and 2x obliquity) in the temperature record.

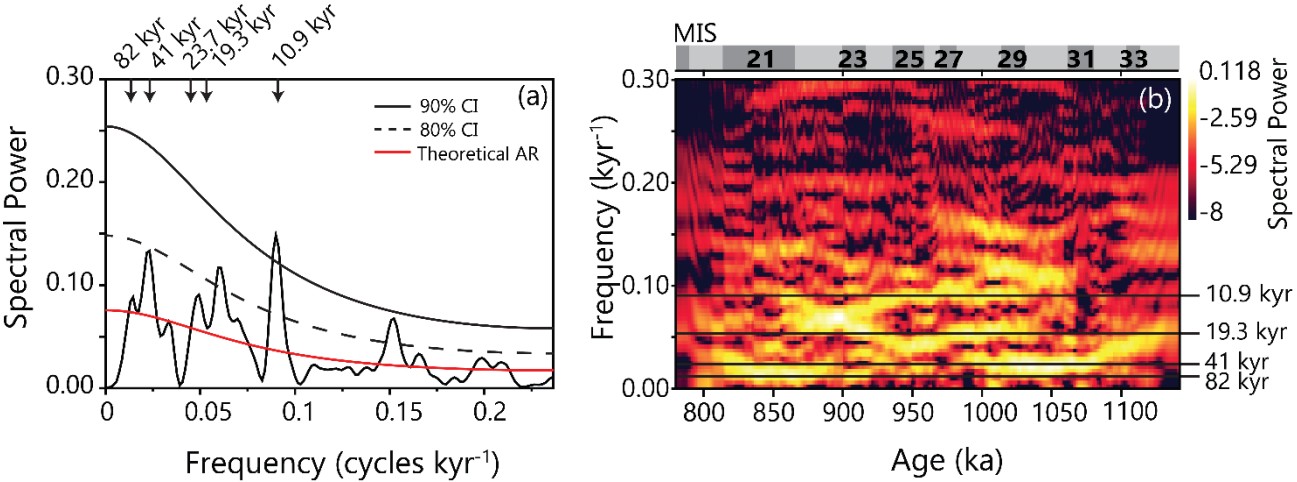

**Figure 4: Frequency analysis of Lake El'gygytgyn MBT′$_{5ME}$ data from 1142 to 752 ka. A) Lomb-Scargle periodogram, with the 80% and 90% confidence levels as well as the theoretical autoregression levels. B) Short-time Fourier transformation showing the evolutionary power spectrum, with the dominant peaks from (A) indicated with horizontal lines. Marine Isotope Stages (MIS) are**
255 **indicated along the top. Both analyses were performed using PAST3 (Hammer et al., 2001).**

## 5. Discussion

### 5.1 Interpretation of the *n*-alkane ACL record

The long-chain *n*-alkanes (*n*-alkanoic acids, *n*-alcohols) with 27 to 35 carbon atoms are biomarkers of terrestrial higher plants (Eglinton and Hamilton, 1967). Indeed, long-chain ($C_{27}$-$C_{33}$) *n*-alkanes are predominantly derived from higher
terrestrial plants in the El'gygytgyn catchment (Wilkie et al., 2013), similar to observations at other Arctic locations (Daniels et al., 2017; O'Connor et al., 2020). Over the MPT study interval, CPI values average 3.6 indicating that higher plant inputs

dominate the *n*-alkane pool (Fig. S1). Both the distributions and isotopic composition of *n*-alkanes can provide information on past vegetation change or the response of vegetation to climate (e.g., temperature or precipitation; Meyers, 2003; Castañeda and Schouten, 2011). We did not examine leaf wax deuterium or carbon isotopes in this study, but we observe changes in the *n*-alkane ACL across the MPT (Fig. 2). Prior studies with independent temperature or aridity reconstructions (e.g., from leaf wax isotopes, lignin phenols, GDGTs or pollen) have reported that ACL increases with increasing aridity (e.g., Liu and Huang, 2005; Schefuß et al., 2003; Peltzer, 1989; Poynter et al., 1989) or increasing temperature (e.g., Castañeda et al., 2009; Zhang et al., 2006; Kawamura et al., 2003; Rommerskirchen et al., 2003; Gagosian and Peltzer, 1986). However, at some locations there is no clear relationship between ACL and temperature or aridity. Furthermore, global correlations are weak hindering a quantitative climate assessment using ACL (Bush and McInerney, 2013). Nonetheless, in the Arctic, higher ACL values have been associated with arid conditions (Andersson et al., 2011). At Lake El'gygytgyn, Keisling et al. (2017) suggested that ACL during the Pliocene is correlated with independent metrics of aridity but is secondarily affected by temperature and vegetation change. As there are available pollen assemblage data from Lake El'gygytgyn spanning the MPT (Zhao et al., 2018), we refine our interpretation of ACL here.

Zhao et al. (2018) examined pollen assemblages in the Lake El'gygytgyn drill core spanning 1,091 to 715 ka and found that shrub tundra and cold steppe communities dominate throughout the study interval. The pollen record exhibits a clear response to glacial-interglacial climate forcing with higher percentages of herbaceous taxa (*Poaceae*, *Cyperaceae*, and *Artemisia*) present during glacial periods, reflecting an open landscape (treeless or shrubless) and cold and arid conditions. Conversely, during interglacials increased percentages of tree and shrub pollen are present, with dwarf birch (*Betula*) and shrub alder (*Alnus*) being the most common (Zhao et al., 2018). The authors developed a landscape openness index, which is the relative difference between the maximum scores of forest and open biomes, as a qualitative assessment of forest vs. an open landscape (Zhao et al., 2018). This record shows higher (positive) values during interglacials, with especially high values noted during MIS 31 and MIS 25 (Fig. 2a). Throughout the MPT, a shift in the landscape openness index is observed with an increasingly open landscape noted after around 890 ka, reflecting a long-term cooling and drying trend during the MPT (Zhao et al., 2018). We find that the ACL at Lake El'gygytgyn tracks changes in the landscape openness index, with lower values observed prior to ~900 ka and higher values afterwards when the landscape is more open and herbaceous taxa are more prevalent (Fig. 2b). Thus, we interpret ACL variations as representing large-scale vegetation changes around Lake El'gygytgyn, which likely reflect a combination of climate-driven vegetation change coupled with the direct response of plants to moisture availability.

## 5.2 Interpretation of the brGDGT record

Prior to interpreting the brGDGT record, the source(s) of the brGDGTs (soil or lacustrine), the seasonality of production, and the choice of temperature calibration need to be considered. The distribution of brGDGTs in our dataset indicates a dominant lacustrine source (Fig. S5) throughout the MPT, in agreement with other previously studied time intervals of the Lake

El'gygytgyn record (Holland et al., 2013; D'Anjou et al., 2013; de Wet et al., 2016; Keisling et al., 2017). Moreover, the samples are dominated by 5-methyl brGDGTs, suggesting that the MBT$'_{5ME}$ index is most suitable for reconstructing temperatures at Lake El'gygytgyn (Fig. S4). There are currently several lacustrine MBT$'_{5ME}$ calibrations that can be applied to reconstruct past temperature (Dang et al., 2018; Russell et al., 2018; Feng et al., 2019; Zhao et al., 2021; Martínez-Sosa et al., 2021; Raberg et al., 2021). Naturally, the reconstructions based on MBT$'_{5ME}$ all show the same temporal structure. The calibration of Dang et al. (2018) is based on alkaline lakes, and so is inappropriate to apply at Lake El'gygytgyn, which has a pH of ~6 (Cremer et al., 2005). Likewise, the calibration of Feng et al. (2019) (Eq. 7) yields unrealistically low temperatures and shows strong deviations from other local and global climate records (Fig. S8) and is deemed not to be applicable here. The calibration of Russell et al. (2018) reconstructs mean annual air temperature (MAAT) and is based on a suite of tropical African lakes. The resulting MAAT from that calibration dramatically over-estimates current MAAT at El'gygytgyn but results are similar to summer (JJA) reconstructed temperatures based on pollen (Melles et al., 2012). The Zhao et al. (2021) calibration reconstructs summer water temperature and is based on settling particulate material from a southern Greenland lake. It should be noted that the Zhao et al. (2021) calibration is currently the only MBT$'_{5ME}$ calibration to water temperature; the other studies calibrate to MAAT or air temperature of months above freezing because in-situ measurements of lake water temperature are often not available. The global "Meth" calibration (Eq. 5) of Raberg et al. (2021) yields values generally similar to those based on MBT$'_{5ME}$ (e.g., Russell et al., 2018; Martínez-Sosa et al., 2021) but some features of the data present in other calibrations (and other Lake E'gygytgyn proxies), such as warmth during MIS 21, do not stand out (Fig. S8). The Raberg et al. (2021) calibrations utilize different subsets of brGDGTs and therefore exhibit some differences compared to calibrations based on MBT$'_{5ME}$ (Fig. S8). The global Bayesian calibration includes globally distributed lakes and is based on the MBT$'_{5ME}$ index (Martínez-Sosa et al., 2021). Both Raberg et al. (2021) and Martínez-Sosa et al., 2021 found that the brGDGT temperature calibrations are strongest when calibrated against warm-season temperatures. As glacial-interglacial structure apparent in the MBT$'_{5ME}$ record, and in the pollen spectra (for intervals where it exists), we therefore base our interpretation on MBT$'_{5ME}$. We further evaluate the choice of calibration by looking at the record across a well-constrained glacial-interglacial cycle during the MPT.

MIS 31 (1.082–1.062 Ma) is widely recognized as an exceptionally warm interglacial period (e.g., Maiorano et al., 2009; Teitler et al., 2015), and there are independent temperature estimates (mean temperature of the warmest month) for Lake El'gygytgyn based on pollen assemblages (Fig. 5; Melles et al., 2012). Previously, de Wet et al., 2016 applied the MBT/CBT calibration of Sun et al. (2011) to examine MIS 31. They found a good agreement between their reconstructed temperatures and pollen-inferred temperature of the warmest month. While the Greenland lakes brGDGT calibration (Zhao et al., 2021) yields similar temperature estimates to the pollen-based reconstruction across MIS 31 (Melles et al., 2012; de Wet et al., 2016), the overall amplitude through the remainder of the study interval is larger. Both the African lakes (Russell et al., 2018) and global Bayesian calibrations (Martínez-Sosa et al., 2021) produce similar values and yield comparatively lower MIS 31 temperatures compared to the pollen estimates, as well as an overall lower amplitude of temperature variability (Fig.

5). Given that Lake El'gygytgyn is a deep and cold lake, and today the shallowest parts of the lake do not exceed 5-6 °C

during summer (Nolan and Brigham-Grette, 2007), a calibration with an overall lower amplitude is likely more realistic. Therefore, throughout the remainder of this discussion we plot our MBT′$_{5ME}$ data using the BayMBT calibration (Martínez-Sosa et al., 2021). While a lacustrine-based brGDGT temperature calibration needs to be applied to Lake El'gygytgyn, it should be noted that our samples display a somewhat different brGDGT distribution compared to other lake datasets (Fig. S5), potentially reflecting a need for a pan-Arctic brGDGT calibration or a site-specific calibration. We therefore place more

emphasis on relative trends in the data (warming and cooling events), which remain robust regardless of the calibration applied. Furthermore, given that several studies now document that brGDGT production in high-latitude lakes peaks during summer, and that both older brGDGT calibrations that were based on the combined methylation and cyclization of brGDGTs, as well as the newer MBT′$_{5ME}$ index show the strongest correlation with growing-season temperatures (Pearson et al., 2011; Sun et al., 2011; Shanahan et al., 2013; Zhao et al., 2021; Miller et al., 2018; Martínez-Sosa et al., 2021), we

interpret relative temperature changes at Lake El'gygytgyn as reflecting conditions during the ice-free summer growing season.

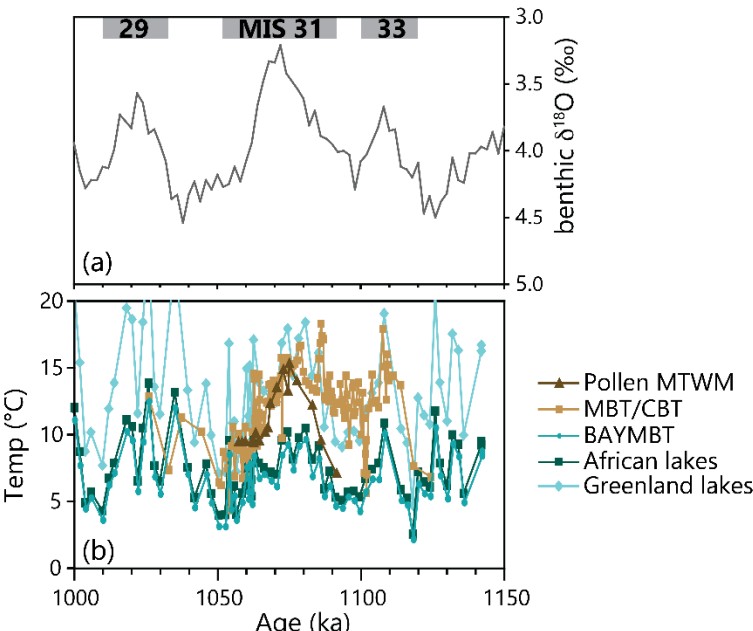

**Figure 5: Comparison of brGDGT calibration during MIS 31 with pollen-derived temperature estimates (Melles et al., 2012). A) The global benthic oxygen isotope stack (Lisiecki and Raymo, 2005) with the different marine isotope stages (MIS) is indicated at**

**the top for reference. B) Lake El'gygytgyn brGDGT temperature estimates using the calibrations of Zhao et al. (2021) based on Greenland lakes, the African lakes MBT′$_{5ME}$ calibration of Russell et al. (2018), the MBT/CBT calibration of Sun et al. (2011) (this is the previously published dataset from de Wet et al. (2016)), the BayMBT calibration (Martínez-Sosa et al., 2021) and pollen mean temperature of the warmest month (MTWM) from Melles et al. (2012). Additional brGDGT calibrations are shown in Fig. S8. Note how the BayMBT and African lakes calibrations yield similar results and nearly plot on top of each other.**

### 5.3 Climate variability during the MPT

Our new brGDGT data from Lake El'gygytgyn document relative temperature changes, revealing several important aspects of orbital and long-term climate variability throughout the MPT. The $MBT'_{5ME}$ record shows that temperatures varied at Milankovitch and sub-Milankovitch time scales, and generally follows changes noted in the global benthic oxygen isotope stack (Lisiecki and Raymo, 2005) and insolation (Fig. 3). This observation agrees with prior studies of Lake El'gygytgyn (e.g., de Wet et al., 2016), although it is evident that the relative importance of obliquity and precession-pacing has varied temporally (Fig. 3 and 4). Our $MBT'_{5ME}$ temperature record supports the observation of Melles et al. (2012) that congruency between peak precession, obliquity, and eccentricity, together with inter-hemispheric teleconnections can generate superinterglacial periods such as MIS 11 and 31. The brGDGT data generally support this interpretation during MIS 31, as well as during the alignment of northern hemisphere summer perihelion and obliquity during the warm MIS 21 (Huybers and Wunsch, 2005), but we note that several other intervals are as warm in the $MBT'_{5ME}$ record despite differences in the orbital configurations. Furthermore, we note that the inverse is also true for temperatures at Lake El'gygytgyn whereby congruent minima in summer insolation, obliquity, and relatively low eccentricity resulted in very cold conditions during MIS 28 and MIS 22 (Fig. 3).

We observe no long-term trend in the $MBT'_{5ME}$ data (Fig. 3), which contrasts with cooling trends observed in the global benthic $\delta^{18}O$ stack (Lisiecki and Raymo, 2005; Clark et al., 2006; Li et al., 2004), Mg/Ca-derived temperatures of Atlantic deep waters (Sosdian and Rosenthal, 2009), and some sea surface temperature records from the northern high latitudes (e.g. Martínez-Garcia et al., 2010; Lawrence et al., 2009; McClymont et al., 2008). At Lake El'gygytgyn, Francke et al. (2013) report changes in orbital frequencies of grain size distributions before and after the MPT, while other proxies at Lake El'gygytgyn do not exhibit any long term trends across the MPT interval (Wennrich et al., 2016). These other proxies do not directly record temperature, yet appear to agree with the brGDGT data in suggesting relatively stable long-term conditions across the MPT. The lack of MPT cooling at El'gygytgyn is difficult to explain given the expansion of northern hemisphere ice sheets at that time. It could imply that the climate at the study site is not representative of the pan-Arctic region, and indeed, there is considerable spatial variability in climate change across the Arctic (Daniels et al., 2021; Tulenko et al., 2020). Alternatively, it may suggest that Arctic cooling was not the critical driver of intensified ice sheet growth, implicating a strong role for the regolith removal hypothesis (Clark and Pollard, 1998; Yehudai et al., 2021) or southern hemisphere (i.e. Antarctic) cooling and ice sheet expansion (Ford and Raymo, 2020). Although we see no overall trend in the brGDGT data, there is a notable increase in leaf wax ACL values reflecting a significant climate-driven ecological change across the MPT (Fig. 2), most likely indicating aridification of the study region. A long-term aridification trend over the past ~1 Myr is similarly noted from the Chinese Loess Plateau (Zhao et al., 2018; Zhou et al., 2018; Wu et al., 2020). At El'gygytgyn, aridity is in part controlled by the amount of moisture being sourced from the nearby high-latitude seas. A sedimentary

record from the Northwind Ridge in the Western Arctic Ocean indicates a transition from seasonal sea ice to perennial sea ice around 1 Ma, which could explain the observed trend toward drier conditions (Dipre et al., 2018). The contrast between

385 enhanced sea ice coverage and stable temperatures at El'gygytgyn points to strong geographic variations in climatic cooling across the MPT, particularly between the marine realm and northeast Russia.

A characteristic feature of the MPT is the appearance of longer glacial cycles, expressed in the frequency domain of many palaeoceanographic and continental records as increasing power in the quasi-100 kyr band, although there is no fundamental

change in orbital forcing across the MPT. In the Arctic, precession and obliquity are key drivers of warm-season temperatures because of the large changes in peak summer insolation and changes in summer duration. Indeed, spectral analysis of the Lake El'gygytgyn MBT′$_{5ME}$ record reveals significant obliquity and precession cycles over the MPT (Fig. 4), with the obliquity signal being the strongest prior to ~1.0 Ma (Fig. 4). Increased variability at longer wavelengths is observed after ~900 ka but at an ~82 kyr cyclicity, which may be twice an obliquity signal rather than eccentricity. However, we note

that the interval studied here may be too short to fully evaluate changes in longer orbital frequencies occurring during the MPT. Prior studies of the Lake El'gygytgyn drill core report similar spectral results but with some notable differences in the relative strength of various frequencies. Grain-size data, reflecting climate-dependent (glacial-interglacial) clastic sedimentation processes at Lake El'gygytgyn, also display a strong obliquity cycle throughout the MPT. Yet in contrast to the brGDGT data, the strongest obliquity signature is noted in the younger part of the record from 950 to 670 ka (Francke et

al., 2013). The grain-size data also exhibits a strong precession cycle from 1100 to 900 ka (Francke et al., 2013), in agreement with the brGDGT data. The authors note that changes in grain-size at Lake El'gygytgyn are not directly coupled to changes in global ice volume.

The effect of MPT ice sheet expansion on spectral signatures would naturally be strongest in direct proximity to where the

405 ice sheets develop, namely North America, Greenland, and Fenno-Scandia. In northeast Siberia, however, ice sheets have not been as prominent a feature of the Pleistocene environment (Brigham-Grette et al., 2013), and it is not entirely clear how the 100 kyr ice sheet influence is transferred to Lake El'gygytgyn. Through model simulations, Melles et al. (2012) indicated that temperature variability at Lake El'gygytgyn is decoupled from Greenland Ice Sheet growth/decay. Instead, precession variations here are more likely connected with regional insolation differences or latitudinal climatic teleconnections (Francke

et al., 2013), which may help explain the lack of brGDGT-inferred cooling across the MPT.

Interestingly, the Lake El'gygytgyn brGDGT data exhibit the strongest spectral power at ~10.9-kyr cyclicity (Fig. 4), which could reflect a half-precession signal (Verschuren et al., 2009). The signal is strongest between 850 and 1030 ka, and is apparent, for example, in the structure of the substages of MIS 21. Sub-orbital climate variations have been noted at Lake

El'gygytgyn but not discussed in detail (e.g. Wennrich et al., 2016), as most proxies are dominated by the Milankovitch frequencies. Nonetheless, half-precession has been observed during the MPT in the northern high latitudes. Haneda et al.

(2020) report half-precession variability in the Kuroshio current of the western North Pacific during MIS 19. Likewise, Ferretti et al. (2010) report that during MIS 21, foraminifera $\delta^{18}O$ varied strongly at the 10.6 kyr wavelength at Site U1313 (Fig. 1; this Site is a revisit of DSDP 607) and a number of other sites across the North Atlantic. Their finding supports a previous identification of Milankovitch harmonics in the North Atlantic (Wara et al., 2002). In that region, the isotope signal was driven by variations in temperature and the strength of Atlantic Meridional Overturning Circulation (AMOC).

The presence of half-precession variability in the North Atlantic has been attributed to non-linear feedbacks due to orbital forcing within the North Atlantic region, related to the different timescales of ice sheet dynamics, deep ocean convection, and moisture feedbacks (Wara et al., 2002). It has alternatively been linked to tropical hemi-precession (e.g. Verschuren et al., 2009) being transmitted to the high latitudes via oceanic and atmospheric teleconnections (Ferretti et al., 2010). In the Northwest Pacific, Haneda et al. (2020) suggest a combination of AMOC variation and equatorial insolation forcing generated a signal of half-precession during MIS 19. Half-precession (9.2-12.7 kyr variance) is also apparent in late-Pleistocene thermocline temperatures from the Western Equatorial Pacific, resulting in sub-orbital variability in the east-west gradient across the equatorial Pacific (Jian et al., 2020). The thermocline water temperature gradient plays a controlling role in the dynamics of the El Niño Southern Oscillation and Walker circulation in the tropical Pacific, which in turn have been linked to climate in the Bering Strait region, mainly through their influence on the position and strength of the Aleutian Low pressure system (Niebauer, 1988). Considering that Bering Strait climatology can exert a strong influence on temperatures at Lake El'gygytgyn (Nolan et al., 2013), this atmospheric teleconnection may be the most direct explanation for the occurrence of half-precession in the El'gygytgyn MBT′$_{5ME}$ record.

In addition to potentially influencing sub-orbital climate dynamics at El'gygytgyn, strengthening of Walker circulation between 1.17 Ma and 0.9 Ma has been hypothesized to be a key driver of the mid-Pleistocene transition (McClymont and Rosell-Melé, 2005). Walker cell intensification likely resulted in a westward shift and deepening of the Aleutian low, thereby lowering air and sea surface temperatures over the Bering Sea and potentially contributing to enhanced upwelling and expanded sea ice (Worne et al., 2021). In contrast to the potential half-precession sensitivity of Lake El'gygytgyn MBT′$_{5ME}$ to tropical/sub-tropical Pacific dynamics, we do not see a clear effect Walker circulation changes across the MPT in the MBT′$_{5ME}$ temperature record presented here, potentially because changing position of the Aleutian low can result in variable temperature responses at El'gygytgyn (Nolan et al., 2013). Furthermore, McClymont and Rosell-Melé (2005) speculate that intensification of Walker circulation increased winter precipitation in the Arctic, contributing to ice sheet expansion. This mechanism contrasts with the vegetation change and moisture reduction across the MPT suggested by pollen and leaf waxes at El'gygytgyn, potentially because of locally expanded sea-ice or lower continental temperatures suppressing evaporation in Lake El'gygytgyn moisture source regions. We note that the half-precession signal in the brGDGT record (Fig. 4) gets stronger between approximately 1.1 and 0.9 Ma, while Walker circulation was intensifying and

450 prior to the first skipped interglacial. Future climate modeling could elucidate how shifts in Walker circulation may have influenced sub-orbital variations in Pacific climate variability, including its influence at Lake El'gygytgyn.

In the following discussion, we take a closer look at glacial-interglacial variability noted in our new Lake El'gygytgyn biomarker records.

### 5.3.1 MIS 34-26 (1150 to 959 ka)

In the interval spanning MIS 34 to 26, we observe a strong correspondence between the MBT$'_{5ME}$ temperatures and global benthic foraminifera oxygen isotope ($\delta^{18}O$) stack (Lisiecki and Raymo, 2005) (Fig. 3). The coldest periods in the MBT$'_{5ME}$ record align closely with glacial Facies A in the El'gygytgyn sediment profile (Fig. 3), as well as with biomarker inferred expansions of sea ice beyond the marginal ice zone of the Bering Sea (Fig. 6; Detlef et al., 2018) indicating cooling across

the Bering region during the MIS 32, 30, 28, and 26. While precession and half-precession are apparent in the evolutionary power spectrum (Fig. 4), the temperature was primarily paced at 41 kyr over this interval, as seen in alignment between the smoothed brGDGT temperature record and the obliquity history (Fig. S9). This could reflect the importance of summer duration on Arctic summer temperature or lake water temperature. Alternatively, the 41-kyr pacing may be driven by changes in glacial/interglacial $p$CO$_2$ variations, for which there is not a well-resolved record over this timespan but it likely

tracked global climate conditions at 41 kyr pacing (Berends et al., 2021).

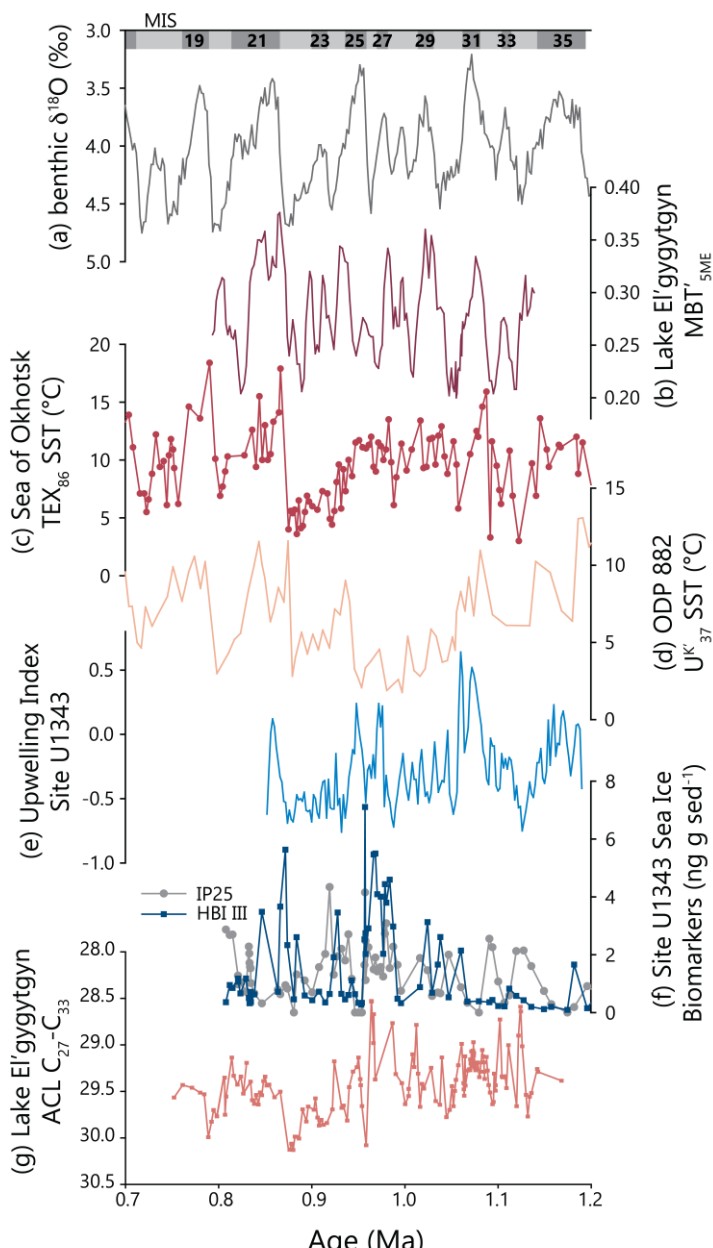

**Figure 6: Comparison of the Lake El'gygytgyn temperature reconstruction across the MPT with marine records from the sub-Arctic North Pacific (for locations of each site, refer to Fig. 1). A) Global benthic oxygen isotope stack (Lisiecki and Raymo, 2005). B) Smoothed (5-point moving average) Lake El'gygytgyn MBT′$_{5ME}$ values. C) TEX$_{86}$ SST from the Sea of Okhotsk (Site MD01-2414; Lattaud et al., 2018). D) U$^{K}$37 SST from the sub-Arctic North Pacific (ODP 882; Martínez-Garcia et al., 2010). E) Upwelling index in the Bering Sea (Site U1343; Worne et al., 2020). F) Concentration of sea ice biomarkers in the Bering Sea (Site U1343; Detlef et al., 2018). G) Lake El'gygytgyn leaf wax ACL values.**

During MIS 31, the MBT$'_{5ME}$ record shows somewhat surprising results in comparison to other Lake El'gygytgyn proxies (Fig. 3). In the El'gygytgyn record, superinterglacial periods (interglacials that are exceptionally warm and wet) were previously identified based on the presence of Facies C, a sedimentary unit characterized by weakly laminated reddish oxidized sediments, high Si/Ti ratios, and low organic matter content (Melles et al., 2012). Facies C is interpreted as reflecting an oxygenated water column, high aquatic productivity, seasonal lake ice, and generally warm conditions (Melles et al., 2012). Pollen spectra from Facies C intervals typically corroborate this interpretation, exhibiting a temporary appearance of birch, alder or other trees (Melles et al., 2012). The MBT$'_{5ME}$ data show a temperature increase of ca. 5-8 °C (using the BayMBT calibration; Martínez-Sosa et al., 2021) compared to the glacial stages immediately preceding and following. However, MIS 29, 27 and 21 were as warm or warmer than MIS 31 in the MBT$'_{5ME}$ record, yet Facies C is absent during these intervals. The differing redox conditions seen in Facies C may have impacted brGDGT distributions, suppressing the warming signal during MIS 31. However, previous empirical evidence suggests that oxic conditions would most likely accentuate MIS 31 warmth (Buckles et al., 2014; Martínez-Sosa and Tierney, 2019), which is not what we observe. We cannot fully exclude other microbial factors that may impact brGDGT producers, but we can infer that other meteorological variables contributed to the presence of Facies C. Based on mineral characteristics, Wei et al. (2014) demonstrated that precipitation is a key driver of Facies C formation. This may imply that MIS 29, 27, and 21 were relatively arid in comparison to MIS 31. The pollen record indicates that cool conifer forest was present around Lake El'gygytgyn during MIS 31 and that higher precipitation characterized this superinterglacial (Zhao et al., 2018; Lozhkin and Anderson, 2013; Melles et al., 2012). Additionally, it is possible that MIS 31 was windier than other interglacials, thereby decreasing water column stability and increasing ventilation and nutrient availability in the surface waters. Indeed, high Si/Ti values, a proxy for diatom productivity, are noted in the Lake El'gygytgyn record during MIS 31 (Melles et al., 2012).

In the interval from MIS 34 to 26, the *n*-alkane ACL record has the highest resolution during MIS 31. At this time, a shift to lower ACL values is noted (Fig. 2 and 5), potentially driven by wetter conditions. During the other interglacials in this interval (MIS 33, 29, and 27), lower ACL values generally occur during peak interglacial conditions. However, low sample resolution during MIS 27 hampers full evaluation of this relationship and higher variability in ACL values is noted during MIS 33, 29 and 27 compared to MIS 31. In the pollen record, MIS 29 is characterized by *Alnus* and *Betula* pollen while MIS 27 is characterized by an increase in larch, dwarf birch and alder pollen (Zhao et al., 2018). Our MBT$'_{5ME}$ record differs from the pollen record in that MIS 29 appears warmer compared to MIS 27 whereas in the pollen record, MIS 27 is interpreted as the warmer and wetter of these interglacials (Zhao et al., 2018). The Lake El'gygytgyn pollen record displays a sharp increase in cold steppe pollen at ~975 ka indicating the onset of glacial MIS 26 (Zhao et al., 2018). This dramatic change appears to be reflected in the ACL record with the largest shift in our record initiating at ~963 ka, when ACL values increase significantly (Fig. 2).

## 5.3.2 MIS 25-22 (959-866 ka)

In the Lake El'gygytgyn brGDGT record, MIS 25 is comparatively cooler than other interglacial periods (Fig. 3). Summer insolation during MIS 25 reaches its highest value of the study interval yet reconstructed temperatures are similar to the preceding and following glacial stages when insolation and obliquity were lower. MBT′$_{5ME}$ values average ~0.25, or 6°C on the BayMBT calibration. Following a brief increase in *n*-alkane ACL values at the termination of MIS 26, ACL again decline during the MIS 25 obliquity maximum circa 953 ka, coinciding with the expansion of trees and shrubs in the area as noted in the pollen record (Zhao et al., 2018). Whereas Zhao et al. (2018) infer warm conditions during MIS 25, the cool brGDGT-inferred temperatures suggest that the vegetation change was rather driven by a combination of longer growing seasons and increase in the moisture balance. During MIS 25, the El'gygytgyn Si/Ti ratio is slightly elevated (Fig. 3), suggesting an increase in lake productivity. In the absence of increased summer temperatures, the change in productivity was likely caused by a combination of longer growing season, increased runoff, or increased windiness promoting mixing of the water column. Regionally, the weak MIS 25 warming agrees well with alkenone-derived SSTs from ODP 882 in the North Pacific (Martínez-Garcia et al., 2010), which also exhibits a relatively cool MIS 25. At Site U1343 in the nearby Bering Sea, palaeoceanographic changes are complex. A low abundance of sea ice biomarkers points to ice-free conditions during MIS 25 (Fig. 6g; Detlef et al., 2018), while high opal accumulation rates (Kim et al., 2014), the presence of ice-marginal diatom species (Worne et al., 2021), and high nutrient upwelling index (Worne et al., 2020) indicate a peak in marginal sea ice conditions, increased wind strength, and a longer sea ice melt season. If these changes in wind strength and seasonality extended over Arctic Asia, it would help explain the limnological and vegetation changes observed at El'gygytgyn.

Numerous MPT studies note a global cooling trend with increasing sea ice extent culminating in an anonymously cool or "skipped" MIS 23 interglacial and a strong glacial period during MIS 22 (Head and Gibbard, 2015 and references therein), in effect giving rise to the first long glacial cycle of the late-Pleistocene. The lack of warming at El'gygytgyn during MIS 25 obscures the nature of the first long glacial cycle from MIS 24-22 that is apparent in the δ$^{18}$O history (Lisiecki and Raymo, 2005; Clark et al., 2006). From MIS 24 to 22, there is a strengthening of sub-orbital temperature variations, with two short-lived warming excursions bracketing a brief cold period at 905 ka despite peaks in both obliquity and northern hemisphere summer insolation (Fig. 3). The evolutionary power spectrum shows that the strength of the 41-kyr climate variability weakens from MIS 25-22 whereas the sub-Milankovitch frequencies become more prominent (Fig. 4), suggesting a breakdown of the climate dynamics that dominated previously. The lower resolution ACL data are in closer agreement with the benthic isotope stack, exhibiting an increasing trend from MIS 25 to 22 and an abrupt decrease at the beginning of MIS 21 (Fig. 3).

Overprinting the increase in ACL values are a series of glacial-interglacial oscillations. For reasons that are not clear, pollen is absent in Lake El'gygytgyn sediments during MIS 23 (Zhao et al., 2018) yet *n*-alkanes are present. During the peak of MIS

23, ACL values are high, and are higher than those of the previous glacials, suggesting conditions that were either arid or more glacial-like (Fig. 2). Temperatures during MIS 22 were particularly cold and arid, characterized by some of the lowest MBT′$_{5ME}$ values and ACL values of the record (Fig. 3). These cool conditions were associated with particularly low productivity, seen in the Si/Ti ratio (Fig. 3). There was a two-phase termination of MIS 22, with an abrupt warming at 880 ka followed by a brief return to cold conditions just prior to the final warming event. The two-phase deglaciation is also apparent in the doublet of Facies A during MIS 22 (Fig. 3). The deglacial warming is approximately synchronous with the increasing obliquity, leading the deglacial decrease in δ$^{18}$O (Fig. 3).

### 5.3.3 MIS 21-20 (866-790 ka)

While MIS 29, 27, and 21 all show MBT′$_{5ME}$ values as high as the superinterglacial MIS 31, the most pronounced warm period of the brGDGT record is MIS 21 (Fig. 3). Warming in the MBT′$_{5ME}$ starts early in MIS 22 and continues to ~868 ka when peak interglacial conditions are noted. Temperatures increased by at least 4 °C and possibly up to 10 °C at the start of MIS 21 as seen in the BayMBT calibration. The comparatively rapid warming recorded at Lake El'gygytgyn from MIS 22 to MIS 21, agrees with Pacific records (Fig. 6; Martínez-Garcia et al., 2010; McClymont et al., 2013; Kender et al., 2018; Lattaud et al., 2019). From 865-845 ka shrubby vegetation communities, including stone pine, birch and alder, returned to the region, while *n*-alkane ACL decreased, thereby suggesting warm and wet conditions (Zhao et al., 2018). The reemergence of tree and shrub communities during MIS 21 is also reflected in the loess sediments of northeast China (not shown; Zhou et al., 2018) suggesting a widespread shift to milder conditions. Despite the notable warmth and a slight increase in aquatic productivity seen in the Si/Ti ratios, MIS 21 is not identified as a superinterglacial interval based on the Si/Ti or lithofacies records (Melles et al., 2012).

Interglacial conditions remained relatively warm for approximately 40 kyr during MIS 21-20. The return to glacial conditions at El'gygytgyn differs notably from the global benthic isotope composite (Lisiecki and Raymo, 2005); whereas the δ$^{18}$O data indicate a gradual cooling or gradual development of ice sheets culminating in full glacial conditions around 814 ka, the brGDGT data indicate an earlier and more rapid cooling to glacial conditions around 835 ka, which was then followed by relatively stable or even warming climate moving into MIS 20 (Fig. 3). This early cooling is also seen in the pollen record, which shows the appearance of open steppe vegetation from 845-810 ka (Zhao et al., 2018). The apparently cool, open-landscape environment in the latter half of MIS 21 occurred despite increasing obliquity, reminiscent of the scenario across MIS 25.

### 5.4 Comparison with North Pacific Marine Records

At present, air temperature anomalies at Lake El'gygytgyn are largely governed by variations in the air masses originating over the sub-Arctic North Pacific, the Bering Sea, and the proximal Arctic Ocean (Nolan and Brigham-Grette, 2007). Based on this, Melles et al. (2012) hypothesized that temperatures during past interglacials were responsive to changes taking place

in the North Pacific Ocean. Specifically, the extremely warm and wet superinterglacials at El'gygytgyn were linked to greater stratification and hence higher SSTs in the North Pacific. The increased oceanic stratification, in turn, was hypothesized to be controlled by southern hemisphere processes, namely reduced Antarctic Bottom Water flow into the Pacific basin. Since that study, several new marine SST and upwelling records spanning the MPT have been developed from the North Pacific allowing for a more direct assessment of this proposed mechanism.

     The MPT temperature history at El'gygytgyn generally resembles marine conditions at glacial-interglacial timescales (Fig. 6). Biomarker records of sea ice expansion at Site U1343 (Fig. 6f) in the Bering Sea (Detlef et al., 2018) correspond to cool glacial stages in the brGDGT record. Likewise, the reconstructed warmth and rapid onset of MIS 21 are also seen in $TEX_{86}$ and alkenone-based temperature records from the Sea of Okhotsk (Lattaud et al., 2019) and ODP Site 882 (Martínez-Garcia

et al., 2010), while cool conditions during MIS 25 are seen at both El'gygytgyn and at ODP Site 882 (Fig. 6b, d). However, the relationship with North Pacific upwelling appears more complex. The strength of upwelling in the Bering Sea during the MPT was reconstructed using biogenic silica accumulation rates and $\delta^{15}N$ offsets between Site U1343 in the Bering Sea and Site 1012 in the tropical North Pacific (Stroynowski et al., 2017; Worne et al., 2020). During MIS 25, the cool temperatures recorded at El'gygytgyn (Fig. 6b) and Site 882 (Fig. 6d) are consistent with a temporary increase in the upwelling index at

Site U1343 (Fig. 6e). In contrast, during MIS 31, strengthened upwelling in the Bering Sea corresponds with warm SST at Site 882 and superinterglacial conditions at El'gygytgyn. The rapid transition into the notably warm MIS 21 coincides with warming seen in the nearby marine records (Martínez-Garcia et al., 2010; Lattaud et al., 2019), and is likewise associated with an increase in Bering Sea upwelling (Fig. 6e; Worne et al., 2020).

A secular decline in upwelling at Site U1343 from MIS 30 to 22 corresponds with declining SSTs and expanding sea ice (Fig. 6e, f). These finding challenge the previous hypothesis of Melles et al. (2012) who suggested that reduced upwelling should cause an increase in temperature in the North Pacific and at El'gygytgyn. To explain this, Worne et al. (2020) suggest that North Pacific cooling and sea ice expansion contributed to a reduction in upwelling via enhanced brine rejection on the Bering shelf and associated expansion of North Pacific Intermediate Waters. Worne et al. (2020) further hypothesize that the

reduced upwelling suppresses $CO_2$ transfer from the deep Pacific to the atmosphere, providing an essential feedback to the global climate during MPT. This process was possibly enhanced by sea level decline (Berends et al., 2021; Elderfield et al., 2012) and the initial closure of the Bering Strait during MIS 23 (Kender et al., 2018). This mechanism is supported by North Pacific upwelling and surface water pH changes during the last glacial termination (Gray et al., 2018; Basak et al., 2018). The Lake El'gygytgyn temperatures exhibit no strong trend associated with the decline in North Pacific upwelling. Rather,

temperatures fluctuated, with minima occurring during both MIS 23, the first "skipped interglacial", and again during MIS 22 (Fig. 3 and 6). The high-frequency variability during MIS 23-21 approximates the half-precession timescales, potentially indicating a strengthening of tropical influences on the high latitudes during this critical transition period of the MPT.

While reduced upwelling did not drive a persistent increase or decrease in temperatures at Lake El'gygytgyn across MIS 30-22, increased *n*-alkane ACL values, especially from MIS 26-22, suggest the continental climate was responsive to changes in the North Pacific (Fig. 6g). Vegetation assemblages at Lake El'gygytgyn suggests a progression of arid, open steppe communities (Zhao et al., 2018). Increased sea ice coverage leading up to and during the 900 ka event at MIS 23 (Fig. 6f) could have suppressed moisture transport to Lake El'gygytgyn. Based on estimates of global sea levels during the MPT, Kender et al. (2018) suggest that the Bering Strait was open during both interglacial and glacial periods prior to MIS 24 then open only during interglacials afterwards. It remains unclear whether changes in sea ice volume and North Pacific circulation around MIS 26 prior to the Bering Strait closure were sufficient to suppress moisture transport at Lake El'gygytgyn or if the Bering Strait might have closed during that glacial. High-resolution sea ice reconstructions from the Arctic Ocean are needed to better evaluate moisture source changes across the MPT.

## 6. Conclusions

The Lake El'gygytgyn brGDGT reconstruction presented here is the only time-continuous record of Arctic continental temperatures spanning the MPT, thereby providing novel insights into the role and the response of high-latitude climate through the mid-Pleistocene. Our new brGDGT record of relative temperature variability captures glacial-interglacial climate fluctuations at Lake El'gygytgyn although the ability of brGDGTs to reconstruct absolute temperature is less clear and a pan-Arctic or site-specific calibration may be needed. Spectral analysis of the brGDGT record, in comparison with marine records, suggests that obliquity, precession, and possibly half-precession are persistent characteristics of high-latitude climate across the MPT. While some locations indicate an overall cooling trend throughout the MPT, this is not observed at Lake El'gygytgyn. We find that MIS 31, which is widely recognized as a particularly warm interglacial, does not exhibit exceptional warmth in the brGDGT record but instead find that MIS 21 was an especially warm interglacial both at Lake El'gygytgyn and throughout much of the North Pacific region. Throughout the MPT, Lake El'gygytgyn pollen and *n*-alkane data exhibit a long-term cooling and drying trend, with a shift to an increasingly open landscape noted after around 900 ka (Zhao et al., 2018). The lack of overall MPT cooling in our terrestrial record contradicts upwelling-driven climate trends observed in North Pacific marine proxies. Having a better understanding of history of the Bering Strait's closure will be critical for resolving the differences between terrestrial and marine climate records in the region. Our new data from Lake El'gygytgyn provide a number of constraints for understanding the evolution of Arctic temperature change across the mid-Pleistocene.

## 7. Data Availability

The biomarker (*n*-alkane and GDGT) data used in this study is archived at the NOAA National Centers for Environmental Information: https://doi.org/10.25921/z73y-mx49

## 8. Acknowledgements

This work was supported by the National Science Foundation (award number 1204087) and a University of Massachusetts Amherst Commonwealth Honors College undergraduate research grant awarded to Kurt Lindberg. We thank Martin Melles, Volker Wennrich, and numerous other international collaborators of the Lake El'gygytgyn Drilling Project. We also thank Anders Noren, Kristina Brady and the staff at LacCore for their assistance and support with numerous large sample requests, as well as John Sweeney and Jeff Salacup at University of Massachusetts for technical support.

## 640 9. Supplement

The supplement to this manuscript contains figures S1 through S9 and supporting text.

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
