# Peer review of "Biomarker Proxy Records of Arctic Climate Change During the Mid-Pleistocene Transition from Lake El'gygytgyn (Far East Russia)"

_Climate of the Past, 2021_

## Referee Comment (RC2)

Lindberg and co-authors report on the well-known and unique Lake E sediment record from the Russian Arctic. Their analysis of brGDGT and n-alkane biomarkers allow new and revised interpretations of past environmental change in terms of precipitation and vegetation across the mid-Pleistocene transition (MPT), a global and enigmatic period of climate change. Overall, the paper is very well written with clear accompanying figures that allow the reader to visualize the text appropriately. Although the authors do explore the utility of the brGDGT proxy using various calibrations and statistical analyses, I offer some suggestions for further data interrogation that I hope will benefit the paper. The fact that this paper provides a rare and continuous terrestrial Arctic paleoclimate record across the MPT naturally amplifies its impact. Collectively, with some revisions, this manuscript will be a highly valuable contribution to both the paleoclimate and brGDGT proxy community. I congratulate the authors on a very nice manuscript – it was a pleasure to read.

***Specific Comments***
L11: Please be consistent with ka or kyr throughout the paper. Same applies for Ma and Myr.

L34-35: Maybe I'm just not familiar with ecology well enough, but it sounds oxymoronic to have cool forests indicative of a warm climate?

L40: Might be worth briefly mentioning what proxies these T and P reconstructions are based on.

L72: You say that MPT lake records are particularly rare. Are there any others? If so, would be worth adding some references, otherwise clarify that Lake E is the only known one!

L84: Can you provide a sentence or two on how the age model is constructed (i.e., what geochronological tools/proxies)? Do you have a sense of how the age model uncertainty (good/bad) effects the timing on your biomarker records? This seems particularly relevant for the spectral analyses.

L95: Please clarify which months you include in summer…JJA?

L96: When do shallower regions reach 5-6 degC…summer? Please specify.

L100: Thermodynamics of the lake? Please specify.

L106: It seems like these 41 re-analyzed samples are a sub-set of the original (de Wet et al., 2016). Was there a strategy for why these ones were chosen rather than re-analyzing the entire dataset?

L109: This sentence is a little misleading because I do not believe de Wet et al. (2016) analyzed n-alkanes. Please rephrase. In addition, and similar to my prior comment, these do not represent all samples de Wet et al. (2016) analyzed. Was there reason behind how many and which samples were re-analyzed for brGDGTs and n-alkanes from the original sample set of de Wet et al. (2016)?

L141: I do not agree with the decision to only use 5-methyl indices. Micro/mesocosm experiments for lake brGDGTs (Martínez-Sosa and Tierney, 2019) in addition to the environmental samples from East Africa (Russell et al., 2018) demonstrate a positive correlation between the abundance of total 6-methyl brGDGT isomers and temperature. A lake record from Iceland also shows some 6-methyl isomers strongly correlated with T inferred from alkenones (Harning et al., 2020). Since we do not have culture experiments to better test which brGDGTs are produced across which environmental gradients, I think it is best to explore the full gamut rather than a subset.

L145: Like my previous comment, I suggest, especially since we do not yet have culture studies for brGDGTs and that there are not that many existing empirical lake calibrations that separate the 5 and 6-methyl isomers, all calibrations be tested. These would include:
Feng et al. (2019)
Harning et al. (2020)
Raberg et al. (2021)

Since they are all also from or include high latitude/altitude locations, they should be just as applicable as the East African and Greenland calibrations and may offer some interesting insights (see later comments).

L158: Is 211 the number of samples you analyzed (127) and those from de Wet (85)? If so, should this number read 212?

L164: CPI should be defined in the methods.

L166: Would it be possible to conduct a change point type analysis here? Otherwise, it seems rather arbitrary to select 900 ka as the boundary. I agree it looks like 900 ka reflects a regime shift, but our eyes can do weird things and I find that statistics help us be more objective.

L224: Other Arctic sites not mentioned suggest this assumption may be less robust (e.g., Dion-Kirschner et al., 2020), where the data show that terrestrial plants also produce a substantial amount of mid-chain plant waxes. I also skimmed the Wilkie paper, and it looks like they only reported on alkanoic acids, which do not necessarily correlate with alkanes as implied here. Might be worth briefly expanding on some of these limitations for your ACL interpretation.

L229: How do these previous studies link ACL and aridity? Modern instrumental calibrations or downcore proxy correlations? I'm not as familiar with using alkanes as a precipitation proxy, so it may help other readers in a similar boat. Is there a physiological mechanism that causes this response to precipitation variability?

L233-234: Same comment as above. What proxies is this aridity condition based on? Please briefly mention.

L249: In re to the visual correlation between ACL and landscape openness index, are there any statistical regression techniques that could be used to support your observation?

L251-252: Since you mention at the end of the previous paragraph that you refine the ACL interpretation for Lake E, does your interpretation differ from the prior studies you discussed? If so, in what way?

L257-259: I don't agree with the argument that dominance of 5-methyls suggests MBT'5Me is most suitable. Transfer functions for other non-biomarker proxies (e.g., foraminifera, chironomids, etc.) can include non-dominant taxa that are important for interpreting changes in T, or whatever the variable of interest is. Therefore, I'd suggest exploring all the available calibrations that separate the isomers, even those that may include 6-methyls, and those that include other calibration approaches, such as stepwise forward selection, as I suggest earlier. I'll also note that Russell et al. (2018) present a SFS calibration that does feature a lower RMSE than their MBT'5Me index calibration's.

L261: Fair to exclude Dang et al. (2018). And if the other calibrations I suggest are also not optimal is some similar way, it could be mentioned here, but each should be systematically evaluated.

L265: Raberg et al. (2021) provide a number of Arctic sites in their modern lake brGDGT calibration.

L270: Yes, naturally because they are all use the same index. I think this is also one reason why it will be interesting to try some other non-MBT-index calibrations to test if these patterns are consistent or not and explore what brGDGTs are key drivers in the indices. In this sense, it may also be interesting to conduct a Pearson correlation matrix as Feng et al. (2019) did to explore the relationship between individual brGDGTs. You may find similarities between other calibration Pearson matrices that could support use (or not) of a certain calibration.

L276-293: This paragraph seems to imply that the pollen-inferred T record is more reliable. Both pollen and brGDGT T proxies have various assumptions and therefore only reflect approximations of past events. I think this paragraph could be rephrased to exclude statements such as "unreasonably large, L280" and "more realistic, L284" and perhaps more objectively compare the 2 T proxy records (pollen and brGDGTs). One of the motivations for this study seems to be producing a rare continuous T record through the MPT. However, if we already have that through pollen, what's the value of brGDGTs here, especially if you think that are not realistic? There are many brGDGT studies that link different distributions to various environmental parameters (e.g., pH, salinity, DO, etc.), which may be a more valuable, or at least supplemental, discussion topic to include.

L286: In Figure S5, it would be interesting to plot up some additional environments as well. Just eyeballing, there appear to be some similarities between brGDGT distributions in Lake E and Svalbard fjord sediments (Dearing Crampton-Flood et al., 2019) and marine sediments (see plots in Xiao et al., 2020), as examples.

L318: Do other qualitative or quantitative climate records from Lake E show a long-term cooling trend that could be compared here?

L321-322: Yes, a lack of MPT cooling at Lake E may be odd, but in addition to climate-driven hypotheses, our still growing knowledge of the brGDGT proxy may also limit or obscure these observations.

L413: Or there is another environmental or microbial factor that contributes to brGDGT distribution changes (e.g., Weber et al., 2018; De Jonge et al., 2019, 2021) and Facies C is still indicative of T.

L440: You explain ACL earlier as a proxy for aridity, so might be better to limit your interpretations of ACL to that here as well. Why would trees expand under increased aridity?

L492: Clarify that this is increased "oceanic" stratification. It took me a few reads to realize you weren't referring to Lake E.

L449: Can you specify which SST proxies? This makes it easier for the reader to independently compare different proxies, as each have different assumptions and interpretations.

Figure 1: Panel B shows 2 yellow dots but only one is mentioned in the caption. Please clarify.

Figure 3: Please clarify in the caption what the bold line is for n-alkane ACL…also a 5-point moving average?

Figure S5: Would be good to include all the lake sediment samples I mention from the other calibration studies in the ternary diagram.

***Technical Comments***
L22: Change "exhibits" to "exhibit". Data are plural

L46: Comma after "ka"

L60: Insert "may have" or something similar before "worked". Otherwise, it sounds like this *is* what happened

L75: Capitalize "arctic", and check if needed elsewhere throughout the ms

L106: I suggest phrasing as "46.77 to 31.06 m" so that it is consistent with the ages you mention just before (i.e., older/deeper to younger/shallower).

L161: Change "they" to "the peaks" or something. It's always easier to follow if pronouns are not used.

L167: Extra and/or missing words around "samples characterized". Please clarify.

L362: Journal should not be referenced in in-text citations.

L389: Does it need to be mentioned or clarified that the LR stack is annual rather than summer biased as you argue the brGDGTs are?

L393-394: Are there any statistical regression analyses that can be used to support this observed alignment between brGDGTs and obliquity?

L412: There are some missing words in the latter half of this sentence.

L539: A little wordy/awkward phrasing…perhaps just "Spectral analysis of the brGDGT record, in comparison with marine records, suggests…"

L506: Add "Sea" after "Bering"

**References (Not included in original text)**

Dearing Crampton-Flood, E., Peterse, F., Sinninghe Damste, J.S., 2019. Production of branched tetraethers in the marine realm: Svalbard fjord sediments revisited. Org. Geochem. 138, 103907.

De Jonge, C., Radujkovic, R., Sigurdsson, B. D., Weedon, J. T., Janssens, I., Peterse, F., 2019. Lipid biomarker temperature proxy responds to abrupt shift in the bacterial community composition in geothermally heated soils. Org. Geochem., 137, 103897.

De Jonge, C., Kuramae, E.E., Radujkovic, D., Weedon, J.T., Janssens, I.A., Peterse, F., 2021. The influence of soil chemistry on branched tetraether lipids in mid- and high latitude soils: Implications for brGDGT-based paleothermometry. Geochim. Cosmochim. Ac. 310, 92-112.

Dion-Kirschner, H., McFarlin, J.M., Masterson, A.L., Axford, Y., Osburn, M.R., 2020. Modern constraints on the sources and climate signals recorded by sedimentary plant waxes in west Greenland. Geochim. Cosmochim. Ac. 286, 336-354.

Feng, X., Zhao, C., D'Andrea, W.J., Liang, J., Zhou, A., Shen, J., 2019. Temperature fluctuations during the Common Era in subtropical southwestern China inferred from brGDGTs in a remote alpine lake. Earth Planet. Sci. Lett. 510, 26–36.

Harning, D.J., Curtin, L., Geirsdóttir, Á., D'Andrea, W.J., Miller, G.H., and Sepúlveda, J., 2020. Lipid biomarkers quantify Holocene summer temperature and ice cap sensitivity in Icelandic lakes. Geophys. Res. Lett. 47, 1–11.

Martínez-Sosa, P., Tierney, J., 2019. Lacustrine brGDGT response to microcosm and mesocosm incubations. Org. Geochem. 127, 12–22.

Weber, Y., Sinninghe Damste, J.S., Zopfi, J., De Jonge, C., Gilli, A., Schubert, C.J., et al., 2018. Redox-dependent niche differentiation provides evidence for multiple bacterial sources of glycerol tetraether lipids in lakes. Proc. Natl Acad. Sci. 115, 10926–10931.

Xiao, W., Wang, Y., Liu, Y., Zhang, X., Shi, L., Xu, Y., 2020. Predominance of hexamethylated 6-methyl branched glycerol dialkyl glycerol tetraethers in the Mariana Trench: source and environmental implication. Biogeosci. 17, 2135-2148.

---

## Author Comment (AC1)

**Response to the review by Referee #1 on the manuscript cp-2021-66 "Biomarker proxy records of Arctic climate change during the Mid-Pleistocene Transition from Lake El'gygytgyn (Far East Russia)**

We thank Dr. Worne for providing a careful review of our manuscript and for the useful suggestions.

***General Comments***

*The MPT is a perplexing component of Quaternary climate change which provides a unique opportunity to understand the interconnectivity and feedbacks between different components of the climate system on orbital and sub-orbital timescales. Lindberg et al. present an interesting dataset from a unique archive to assess Arctic response to MPT climate dynamics and provide thorough discussion on the role of the North Pacific, Bering Sea, and inter-hemispheric teleconnections which could result in the observed changes in vegetation and temperature in north-eastern Russia. They properly outline the limitations of their datasets and calibrations, interpreting their data to a suitable resolution and using statistically robust techniques. The study concludes that the warm interglacial conditions widely observed during MIS 31, as well as subsequent MPT cooling, are not apparent at Lake El'gygytgyn. They discuss potential causes for this, as well as the detected sub-orbital cyclicity in the temperature record, including the interplay between temperature and moisture availability resulting from teleconnection with Arctic and sub-Arctic, as well as tropical and Atlantic Ocean feedbacks. Overall, I recommend that this manuscript be accepted subject to minor revisions. I have outlined some areas which I feel could benefit from additional discussion, as well as some technical corrections. I hope that the authors find these recommendations helpful.*

***Specific Comments***

***Review:*** *Could a short discussion be made on the difference between the results from the original and re-analysed samples from de Wet et al. (2016)?*

**Reply:** Yes, we can include a short discussion of this in the supplement. As the samples in this part of the core have low amounts of 6-methyl isomers, re-analyzing the samples with the newer HPLC methods yielded results similar to the original ones. Indeed, there is a very strong correlation between the original de Wet MBT values and the MBT values of the re-analyzed samples, with a slope near 1.

That said, the calibration that de Wet et al. (2016) applied (from Sun et al., 2011) was based on the combination of MBT and CBT values, and was also based on a smaller set of lakes than the new BayMBT calibration. While there is a good correlation between the reanalyzed BayMBT-based temperature values and the original MBT/CBT-based values, the slope of these data is approximately 0.5, indicating that that amplitude of temperature variability based on the BayMBT calibration is approximately half that of the MBT/CBT-based calibration.

***Review:*** *Figure 5 presents an MBT/CBT calibration from Sun et al. (2011) in light brown squares, however there appears to be no mention of this calibration is made in the text. There is clearly an offset in values in the MBT/CBT record compared to the African lakes and BAYMBT calibrations at ~1.1 ma. Could the author include some information on this calibration and the likely reason for this discrepancy, as they have for the Greenland lakes calibration? This is particularly relevant for the later discussion on MIS 31 as the MBT/CBT calibration has notably higher temperatures which would support superinterglacial warmth,*

compared to the BAYMBT and African Lakes which do not show a similar peak in temperature or subsequent cooling trend.

**Reply:** We agree with the reviewer that, based on the BayMBT temperature reconstruction, MIS 31 does not stand out as such a strong interglacial as it does based on the MBT/CBT-based reconstruction. The reason for including the MBT/CBT calibration is this is the calibration that de Wet et al. (2016) published their brGDGT data on. Overall, as this calibration is based on the older HPLC methods that did not separate the 5-methyl and 6-methyl isomers, it is not the best one to apply currently. We will add some information to the text about this calibration and its offset from the others.

*Review:*  *The authors include a very interesting discussion on suborbital cyclicity at ~11 kyr. Some mention is given to the potential control of the monsoon. I wonder if the authors has considered the amplification of the Walker Circulation as a mechanism of suborbital cyclicity? Intensification of the Walker Circulation is suggested to have occurred in the build up to the MPT from ~1.17 Ma, propagating to the high latitudes through changes in the El Nino Southern Oscillation and East Asian Winter Monsoon (McClymont & Rosell-Melé, 2005; Stroynowski et al., 2017). Evidence from the adjacent Bering Sea supports this, where sea ice is suggested to have responded to the resultant changes in wind strength, temperature and moisture delivery (Stroynowski et al., 2017; Worne et al., 2021). This would also fit with the authors discussion of changing wind strength and wetter conditions.*

**Reply:** Thank you for this suggestion, we had not seen the Worne et al. (2021) paper yet. We will incorporate the idea that Walker Circulation may have played a role in initiating the changes around 1 Ma into our discussion and also will include the suggested additional citations.

*Review:*  *Line 437: Evidence from the Site U1343 does not show reduced diatom productivity, where the opal MAR record (Kim et al., 2014) is high through this interglacial, indicating high productivity. Furthermore, Detlef et al. (2018) states that "beginning at MIS 25, [Site U1343] is characterised by an ice-free eastern Bering Sea". Recent diatom data may be in better support of your discussion here, where fossil assemblages suggests that MIS 25 represents an interval of peak marginal sea ice conditions across the MPT interval, suggested to be a result of increased wind strength and longer sea ice melt seasons (Worne et al., 2021).*

**Reply:** The climatic and oceanic changes that took place during Marine Isotope Stage 25 are complex, and we thank the reviewer for pointing out the need to improve the discussion around Site U1343 at that time. Indeed, high opal accumulation rates at Site U1343, together with the low sea-ice biomarkers may indicate highly productive, ice free conditions. On the other hand, as the reviewer points out, recent results of Worne et al. (2021), based on diatom assemblages at Site U1343, suggest an increase in marginal sea-ice conditions, possibly related to a lengthened sea-ice melt season. We will revise this paragraph to better characterize the changes going on in the N. Pacific and how they may relate to temperature change at Lake El'gygytgyn.

**Technical Corrections**

Thank you for catching these technical errors. We will make all the corrections listed below.

*Line 12: comma after "Arctic"*

*Line 35: comma after "(Brigham-Grette et al., 2013)"*

*Line 36: rephrase, perhaps to "tundra vegetation, where the tree line lies..."*

*Line 67 and 75: capitalise Arctic*

*Line 114 and 116: extra space before methanol needs removing.*

*Section 3.1: be consistent with capitalisation of Eq or eq.*

*Figure 3: Caption for E and F appears to have errors with references in the wrong place and text missing, perhaps because of referencing software. Needs to be re-written.*

*Line 334: NE has been fully written as northeast earlier in the text, needs to be consistent.*

*Line 450: "short-lived" needs hyphenating.*

**References cited here, not included in original manuscript:**

We will consider these references in our revisions, particularly in the revised discussions on sub-orbital variability and N. Pacific climatic/oceanographic changes. Thank you for bringing them to our attention.

Kim, S., Takahashi, K., Khim, B. K., Kanematsu, Y., Asahi, H., & Ravelo, A. C. (2014). Biogenic opal production changes during the Mid-Pleistocene Transition in the Bering Sea (IODP Expedition 323 Site U1343). Quaternary Research, 81(1), 151–157. https://doi.org/10.1016/j.yqres.2013.10.001

McClymont, E. L., & Rosell-Melé, A. (2005). Links between the onset of modern Walker circulation and the mid-Pleistocene climate transition. Geology, 33(5), 389–392. https://doi.org/10.1130/G21292.1

Stroynowski, Z., Abrantes, F., & Bruno, E. (2017). The response of the Bering Sea Gateway during the Mid-Pleistocene Transition. Palaeogeography, Palaeoclimatology, Palaeoecology, 485(March), 974–985. https://doi.org/10.1016/j.palaeo.2017.08.023

Worne, S., Stroynowski, Z., Kender, S., & Swann, G. E. A. (2021). Sea-ice response to climate change in the Bering Sea during the Mid-Pleistocene Transition. Quaternary Science Reviews, 259, 106918. https://doi.org/10.1016/j.quascirev.2021.106918

Citation: https://doi.org/10.5194/cp-2021-66-RC1

---

## Author Comment (AC2)

**Response to the review by Referee #2 on the manuscript cp-2021-66 "Biomarker proxy records of Arctic climate change during the Mid-Pleistocene Transition from Lake El'gygytgyn (Far East Russia)**

We thank the reviewer for providing a careful review of our manuscript and will incorporate their suggestions, as detailed below.

*Lindberg and co-authors report on the well-known and unique Lake E sediment record from the Russian Arctic. Their analysis of brGDGT and n-alkane biomarkers allow new and revised interpretations of past environmental change in terms of precipitation and vegetation across the mid-Pleistocene transition (MPT), a global and enigmatic period of climate change. Overall, the paper is very well written with clear accompanying figures that allow the reader to visualize the text appropriately. Although the authors do explore the utility of the brGDGT proxy using various calibrations and statistical analyses, I offer some suggestions for further data interrogation that I hope will benefit the paper. The fact that this paper provides a rare and continuous terrestrial Arctic paleoclimate record across the MPT naturally amplifies its impact. Collectively, with some revisions, this manuscript will be a highly valuable contribution to both the paleoclimate and brGDGT proxy community. I congratulate the authors on a very nice manuscript – it was a pleasure to read.*

**Specific Comments**

*Review:* L11: Please be consistent with ka or kyr throughout the paper. Same applies for Ma and Myr.

**Reply:** Following the recommendations in "*Terminology of geological time: Establishment of a community standard*" by Aubrey et al. 2009 (Stratigraphy, vol. 6, no. 2), we use "ka" and "Ma" to represent geohistorical dates while "kyr" and "Myr" and used to represent geohistorical duration. We will carefully check that all uses of ka/kyr and Ma/Myr are correct throughout the manuscript.

*Review:* L34-35: Maybe I'm just not familiar with ecology well enough, but it sounds oxymoronic to have cool forests indicative of a warm climate?

**Reply:** Good point - we understand why this might be confusing. Presently, Lake El'gygytgyn is surrounded by tundra with the nearest trees ~150 km to the south. Thus, having any trees around the lake, even those representing cool forest species, indicates a significantly warmer climate. In our revision, we will re-word this for clarity.

*Review:* L40: Might be worth briefly mentioning what proxies these T and P reconstructions are based on.

**Reply:** The previous temperature and precipitation reconstructions, which only represent short segments of the overall Lake El'gygytgyn record, are based on pollen assemblages. We will add this to the revision.

*Review:* L72: You say that MPT lake records are particularly rare. Are there any others? If so, would be worth adding some references, otherwise clarify that Lake E is the only known one!

**Reply:** There are only a handful of lacustrine records continuously spanning the MPT, including Lake Baikal, Lake Malawi and Lake Ohrid, as only a relatively small number of lakes have been drilled to date.

We will revise this sentence to include appropriate citations and clarify that El'gygytgyn is the only such record from the Arctic.

*Review: L84: Can you provide a sentence or two on how the age model is constructed (i.e., what geochronological tools/proxies)? Do you have a sense of how the age model uncertainty (good/bad) effects the timing on your biomarker records? This seems particularly relevant for the spectral analyses.*

**Reply:** We utilize the age model of Nowaczyk et al. (2013), which is based on iterative tie-point identification using 1) paleomagnetic reversals, 2) comparison of biogenic silica to the LR04 oxygen isotope stack, and lastly 3) comparison of TOC and magnetic susceptibility to summer insolation. Given the resolution and uncertainty of the target curves (LR04 and insolation), the absolute age uncertainty could be off by 3-15 kyr, although the relative age assignments have a precision of 500 years (Nowaczyk et al. 2013). Since the Lake E age model is tuned to LR04, it may be circular to discuss leads and lags between datasets, and this is generally avoided in our discussion. Given the 3$^{rd}$-order tie points improve the relative precision to ~500 years, it remains useful to discuss the spectral results because the multitude of Lake El'gygytgyn proxies appear to exhibit different orbital-scale patterns. We will add a brief summary of these main points to the text about the age model.

*Review: L95: Please clarify which months you include in summer…JJA?*

**Reply:** Yes, we are referring to JJA for summer temperatures. We will specify this in the text.

*Review: L96: When do shallower regions reach 5-6 degC…summer? Please specify.*

**Reply:** Shallower regions within Lake El'gygytgyn reach 5-6 degrees C during the summer. We will clarify this in the text.

*Review: L100: Thermodynamics of the lake? Please specify.*

**Reply:** In this sentence, we are referring to atmospheric temperatures at Lake El'gygytgyn. Nolan et al. (2013) analyzed how atmospheric pressure patterns may or may not relate to the temperature at Lake El'gygytgyn. They found that recent warming signal at El'gygytgyn was not attributed to changes in the frequency of different weather patterns, implying that general warming of the atmosphere (rather than storm tracks) controls air temperature. We will revise this sentence for clarity.

*Review: L106: It seems like these 41 re-analyzed samples are a sub-set of the original (de Wet et al., 2016).*
*Was there a strategy for why these ones were chosen rather than re-analyzing the entire dataset?*

**Reply:** Yes, the re-analyzed samples are a subset of the original 174 samples that de Wet at al. (2016) analyzed. We did not re-run the full dataset due to the cost. Furthermore, as these samples do not contain significant amounts of the 6-methyl isomers, re-analyzing yields similar results. Every 3rd sample was re-analyzed. We will add some text clarifying how we selected samples for re-analysis and will do a better job emphasizing that the re-analyzed samples yielded similar results (we can add a figure to the supplement, also addressing the question raised by the other reviewer). A complete re-analysis of the de Wet samples is not needed, as the higher-resolution sampling is not necessary for the orbital-scale questions investigated here.

**Review:** *L109: This sentence is a little misleading because I do not believe de Wet et al. (2016) analyzed n-alkanes. Please rephrase. In addition, and similar to my prior comment, these do not represent all samples de Wet et al. (2016) analyzed. Was there reason behind how many and which samples were re-analyzed for brGDGTs and n-alkanes from the original sample set of de Wet et al. (2016)?*

**Reply:** The doctoral thesis of de Wet (2017) does include *n*-alkane analyses. This is cited correctly. The de Wet et al. (2016) EPSL publication only includes GDGT data. de wet (2017) ran 85 samples for *n*-alkanes, which we supplement with the 127 samples that we newly extracted for this study.

**Review:** *L141: I do not agree with the decision to only use 5-methyl indices. Micro/mesocosm experiments for lake brGDGTs (Martínez-Sosa and Tierney, 2019) in addition to the environmental samples from East Africa (Russell et al., 2018) demonstrate a positive correlation between the abundance of total 6-methyl brGDGT isomers and temperature. A lake record from Iceland also shows some 6-methyl isomers strongly correlated with T inferred from alkenones (Harning et al., 2020). Since we do not have culture experiments to better test which brGDGTs are produced across which environmental gradients, I think it is best to explore the full gamut rather than a subset.*

**Reply:** In this portion of the El'gygytgyn record, the 6-methyl isomers average 9% of the total brGDGTs. However, we note that many Lake El'gygytgyn samples do not contain any 6-methyl isomers, particularly in the Late Pleistocene. Thus, when analyzing the entire 3.6 Ma record (which we are doing in other publications), we need to use the 5-methyl indices, and so focus on those indices. We can explore use of some of the 6-methyl indices and the calibrations listed below in our revision.

**Review:** *L145: Like my previous comment, I suggest, especially since we do not yet have culture studies for brGDGTs and that there are not that many existing empirical lake calibrations that separate the 5 and 6-methyl isomers, all calibrations be tested. These would include:*
*Feng et al. (2019)*
*Harning et al. (2020)*
*Raberg et al. (2021)*
*Since they are all also from or include high latitude/altitude locations, they should be just as applicable as the East African and Greenland calibrations and may offer some interesting insights (see later comments).*

**Reply:** We will add these calibrations in our revisions.

**Review:** *L158: Is 211 the number of samples you analyzed (127) and those from de Wet (85)? If so, should this number read 212?*

**Reply:** Yes, thank you for catching that. We will correct this in the revised text.

**Review:** *L164: CPI should be defined in the methods.*

**Reply:** The CPI equation is defined in the supplemental text. The reference in the methods will be adjusted to properly reflect that.

**Review:** *L166: Would it be possible to conduct a change point type analysis here? Otherwise, it seems*

*rather arbitrary to select 900 ka as the boundary. I agree it looks like 900 ka reflects a regime shift, but our eyes can do weird things and I find that statistics help us be more objective.*

**Reply:** We conducted a change point analysis on our *n*-alkane data and found the significant ACL shift to occur between 960 and 930ka (95% CI: 1019-895 ka). We retain the use of 900ka as a general partition because that is when numerous marine records in the North Pacific & North Atlantic observe a shift in ice sheet dynamics towards lower frequency variability (Head & Gibbard 2015). However, 900 ka is still only an estimated average for the timing, globally, therefore we believe that our analysis of ACL before and after 900 ka to still be valid. We will add a description of the change-point analysis and its results.

[Figure]

**Review:** *L224: Other Arctic sites not mentioned suggest this assumption may be less robust (e.g., Dion-Kirschner et al., 2020), where the data show that terrestrial plants also produce a substantial amount of mid-chain plant waxes. I also skimmed the Wilkie paper, and it looks like they only reported on alkanoic acids, which do not necessarily correlate with alkanes as implied here. Might be worth briefly expanding on some of these limitations for your ACL interpretation.*

**Reply:** We have analyzed both *n*-alkanoic acids and *n*-alkanes on Lake El'gygytgyn samples (this work was conducted by a recent PhD graduate of UMass and in presently in-preparation for publication) and find similar results for both downcore concentrations and isotopic values for both compound classes. The $C_{27}$ *n*-alkane and the $C_{26}$ *n*-alkanoic acid are the most abundant. We still think, given the available evidence, that ACL is generally reflecting the large-scale vegetation changes occurring in the Lake El'gygytgyn catchment.

**Review:** *L229: How do these previous studies link ACL and aridity? Modern instrumental calibrations or downcore proxy correlations? I'm not as familiar with using alkanes as a precipitation proxy, so it may help other readers in a similar boat. Is there a physiological mechanism that causes this response to precipitation variability?*

**Reply:** It has been widely reported in the literature that in samples from certain locations, ACL tracks either changes in temperature or aridity (with longer chain-lengths associated with increased temperature or increased aridity). However, to our knowledge, none of these studies were using ACL alone as the

interpretation for aridity. Rather, typically another proxy is used (pollen, leaf wax isotopes) to provide the aridity interpretation and it was noted at ACL tracked changes in aridity. We will revise this section for additional clarity, including adding an additional citation to Bush & McInerney (2013), who discuss ACL in detail.

*Review: L233-234: Same comment as above. What proxies is this aridity condition based on? Please briefly mention.*

**Reply:** We will clarify that these prior studies based the aridity interpretations on proxies other than ACL.

*Review: L249: In re to the visual correlation between ACL and landscape openness index, are there any statistical regression techniques that could be used to support your observation?*

**Reply:** We have now performed regression analysis between ACL and landscape openness, confirming our qualitative description of how these two variables are negatively correlated. We recognize that a visual presentation of the correlation would be useful, and will include this regression in the supplement.

[Figure]

*Review: L251-252: Since you mention at the end of the previous paragraph that you refine the ACL interpretation for Lake E, does your interpretation differ from the prior studies you discussed? If so, in what way?*

**Reply:** While some prior studies have noted ACL tracks changes in temperature or aridity, we think most authors would agree that this is an oversimplification and multiple factors are responsible. We note that many prior studies do not have both pollen and leaf wax data. In this study, the addition of pollen data shows us that ACL broadly tracks the landscape openness index.

*Review: L257-259: I don't agree with the argument that dominance of 5-methyls suggests MBT'5Me is most suitable. Transfer functions for other non-biomarker proxies (e.g., foraminifera, chironomids, etc.) can include non-dominant taxa that are important for interpreting changes in T, or whatever the variable of interest is. Therefore, I'd suggest exploring all the available calibrations that separate the isomers, even those that may include 6-methyls, and those that include other calibration approaches, such as stepwise forward selection, as I suggest earlier. I'll also note that Russell et al. (2018) present a SFS calibration that does feature a lower RMSE than*

*their MBT'5Me index calibration's.*
**Reply:** We cannot apply the 6-methyl brGDGT indices to some parts of the Lake El'gygytgyn record as only 5-methyl brGDGTs are present. In order to examine the full record (we are exploring this in other publications) and to get a continuous temperature reconstruction, we need to use 5-methyl brGDGTs. We note that the SFS calibration of Russell et al. (2018) includes brGDGT IIb' but as this brGDGT is typically not present in Lake El'gygytgyn samples, we did not apply this calibration.

*Review: L261: Fair to exclude Dang et al. (2018). And if the other calibrations I suggest are also not optimal*
*is some similar way, it could be mentioned here, but each should be systematically evaluated.*

**Reply:** We will evaluate each additional calibration suggested and will include them if deemed valid for comparison.

*Review: L265: Raberg et al. (2021) provide a number of Arctic sites in their modern lake brGDGT calibration.*

**Reply:** We will include the Raberg et al (2021) calibration in our revision as well as the additional recommended calibrations.

*Review: L270: Yes, naturally because they are all use the same index. I think this is also one reason why it will be interesting to try some other non-MBT-index calibrations to test if these patterns are consistent or not and explore what brGDGTs are key drivers in the indices. In this sense, it may also be interesting to conduct a Pearson correlation matrix as Feng et al. (2019) did to explore the relationship between individual brGDGTs. You may find similarities between other calibration Pearson matrices that could support use (or not) of a certain calibration.*

**Reply:** We will investigate this further and, if found to be important, we can add it to the revised manuscript.

*Review: L276-293: This paragraph seems to imply that the pollen-inferred T record is more reliable. Both pollen and brGDGT T proxies have various assumptions and therefore only reflect approximations of past events. I think this paragraph could be rephrased to exclude statements such as "unreasonably large, L280" and "more realistic, L284" and perhaps more objectively compare the 2 T proxy records (pollen and brGDGTs). One of the motivations for this study seems to be producing a rare continuous T record through the MPT. However, if we already have that through pollen, what's the value of brGDGTs here, especially if you think that are not realistic? There are many brGDGT studies that link different distributions to various environmental parameters (e.g., pH, salinity, DO, etc.), which may be a more valuable, or at least supplemental, discussion topic to include.*

**Reply:** We can revise these sentences to include more objective wording. We note that that pollen data only exist for certain portions of the Lake El'gygytgyn record; for much of the Lake El'gygytgyn core, pollen analyses are still in progress. One advantage of brGDGTs, is that the method is faster compared to pollen counting. We note that the pollen data spanning the MPT provides a landscape openness index but the authors did not provide a pollen-based temperature reconstruction (although pollen-based temperature estimates are available for certain discrete parts of the record, such as MIS 5 and MIS 11 (Melles et al., 2012)). Therefore, brGDGTs provide the first continuous temperature reconstruction from

Lake El'gygytgyn spanning the MPT. While, for the reasons described in the text, we caution readers against interpreting the absolute temperature values yielded by brGDGTs, we believe the warming and cooling trends in these data are robust. Certainly, other environmental factors are likely influencing the brGDGT record. We can add some text on this to the supplement.

*Review: L286: In Figure S5, it would be interesting to plot up some additional environments as well. Just eyeballing, there appear to be some similarities between brGDGT distributions in Lake E and Svalbard fjord sediments (Dearing Crampton-Flood et al., 2019) and marine sediments (see plots in Xiao et al., 2020), as examples.*

**Reply:** As Lake El'gygytgyn is a freshwater lake and has been a freshwater lake throughout its entire history, we feel it is appropriate to limit our comparison to other lacustrine datasets.

*Review: L318: Do other qualitative or quantitative climate records from Lake E show a long-term cooling trend that could be compared here?*

**Reply:** One of the novel aspects of our study is that it is one of the first continuous temperature reconstructions from Lake El'gygytgyn. There are numerous other proxies from this core spanning the MPT. While none show a clear cooling during the MPT, we also note that none are direct temperature proxies. We can add some information on other Lake El'gygytgyn proxies in our revision.

*Review: L321-322: Yes, a lack of MPT cooling at Lake E may be odd, but in addition to climate-driven hypotheses, our still growing knowledge of the brGDGT proxy may also limit or obscure these observations.*

**Reply:** We agree and feel that we have clearly noted the limitations of brGDGT data throughout the manuscript.

*Review: L413: Or there is another environmental or microbial factor that contributes to brGDGT distribution changes (e.g., Weber et al., 2018; De Jonge et al., 2019, 2021) and Facies C is still indicative of T.*

**Reply:** It is certainly possible that another factor contributes to brGDGT distribution changes and we can mention this in the revised manuscript. One advantage of using the BAYMBT calibration is that it spans a wide range of environments.

*Review: L440: You explain ACL earlier as a proxy for aridity, so might be better to limit your interpretations of ACL to that here as well. Why would trees expand under increased aridity?*

**Reply:** Based on the Lake El'gygytgyn pollen record published in Zhao et al. (2018), the amount of tree and shrub pollen increases during MIS 25, which is paired with a decrease in *n*-alkane ACL at this time. Based on our interpretation of ACL increasing with aridity and cooler brGDGT-inferred temperatures, this decrease observed during MIS 25 suggest wetter conditions in the region that would support the expansion of trees and shrubs. We will revise this sentence to read indicate that the decline in ACL coincides with an expansion of trees and shrubs in the pollen record.

*Review: L492: Clarify that this is increased "oceanic" stratification. It took me a few reads to realize you weren't referring to Lake E.*

**Reply:** We will make this change.

*Review:* L449: Can you specify which SST proxies? This makes it easier for the reader to independently compare different proxies, as each have different assumptions and interpretations.

**Reply:** Yes, we will specify which SST proxies are used in the N. Pacific paleoceanographic records. These include alkenones, diatom assemblages and opal mass accumulation rates, and nitrogen-isotope records of nutrient utilization.

*Review:* Figure 1: Panel B shows 2 yellow dots but only one is mentioned in the caption. Please clarify.

**Reply:** We will revise the figure to only include the location of the drill core 5011-1 analyzed in this study. This figure was previously re-used from a different study that also included a second core, piston core LZ1024, which spans the past ~300 ka. Thank you for catching this.

*Review:* Figure 3: Please clarify in the caption what the bold line is for n-alkane ACL...also a 5-point moving
Average?

**Reply:** We will clarify this in the caption. Yes, it is a 5-point running mean.

*Review:* Figure S5: Would be good to include all the lake sediment samples I mention from the other calibration studies in the ternary diagram.

**Reply:** We can include samples from other calibration studies that are publicly available.

**Technical Comments**

Thank you for catching these errors. We will make these corrections.

*L22: Change "exhibits" to "exhibit". Data are plural*

*L46: Comma after "ka"*

*L60: Insert "may have" or something similar before "worked". Otherwise, it sounds like this is what happened*

*L75: Capitalize "arctic", and check if needed elsewhere throughout the ms*

*L106: I suggest phrasing as "46.77 to 31.06 m" so that it is consistent with the ages you mention just before (i.e., older/deeper to younger/shallower).*

*L161: Change "they" to "the peaks" or something. It's always easier to follow if pronouns are not used.*

*L167: Extra and/or missing words around "samples characterized". Please clarify.*

*L362: Journal should not be referenced in in-text citations.*

*L389: Does it need to be mentioned or clarified that the LR stack is annual rather than summer biased as you argue the brGDGTs are?*

*L393-394: Are there any statistical regression analyses that can be used to support this observed alignment between brGDGTs and obliquity?*

**Reply:** We argue that brGDGTs at Lake El'gygytgyn are aligned with obliquity because the spectral analysis of our record shows a strong 41 kyr signal (Fig. 4a) in line with the period of obliquity.

*L412: There are some missing words in the latter half of this sentence.*

*L539: A little wordy/awkward phrasing…perhaps just "Spectral analysis of the brGDGT record, in comparison with marine records, suggests…"*

*L506: Add "Sea" after "Bering"*

**References (Not included in original text)**

We will consider these references in our revisions.

Dearing Crampton-Flood, E., Peterse, F., Sinninghe Damste, J.S., 2019. Production of branched tetraethers in the marine realm: Svalbard fjord sediments revisited. Org. Geochem. 138, 103907.

De Jonge, C., Radujkovic, R., Sigurdsson, B. D., Weedon, J. T., Janssens, I., Peterse, F., 2019. Lipid biomarker temperature proxy responds to abrupt shift in the bacterial community composition in geothermally heated soils. Org. Geochem., 137, 103897.

De Jonge, C., Kuramae, E.E., Radujkovic, D., Weedon, J.T., Janssens, I.A., Peterse, F., 2021. The influence of soil chemistry on branched tetraether lipids in mid- and high latitude soils: Implications for brGDGT-based paleothermometry. Geochim. Cosmochim. Ac. 310, 92-112.

Dion-Kirschner, H., McFarlin, J.M., Masterson, A.L., Axford, Y., Osburn, M.R., 2020. Modern constraints on the sources and climate signals recorded by sedimentary plant waxes in west Greenland. Geochim. Cosmochim. Ac. 286, 336-354.

Feng, X., Zhao, C., D'Andrea, W.J., Liang, J., Zhou, A., Shen, J., 2019. Temperature fluctuations during the Common Era in subtropical southwestern China inferred from brGDGTs in a remote alpine lake. Earth Planet. Sci. Lett. 510, 26–36.

Harning, D.J., Curtin, L., Geirsdóttir, Á., D'Andrea, W.J., Miller, G.H., and Sepúlveda, J., 2020. Lipid biomarkers quantify Holocene summer temperature and ice cap sensitivity in Icelandic lakes. Geophys. Res. Lett. 47, 1–11.

Martínez-Sosa, P., Tierney, J., 2019. Lacustrine brGDGT response to microcosm and mesocosm incubations. Org. Geochem. 127, 12–22.

Weber, Y., Sinninghe Damste, J.S., Zopfi, J., De Jonge, C., Gilli, A., Schubert, C.J., et al., 2018. Redox-dependent niche differentiation provides evidence for multiple bacterial sources of glycerol tetraether lipids in lakes. Proc. Natl Acad. Sci. 115, 10926–10931.

Xiao, W., Wang, Y., Liu, Y., Zhang, X., Shi, L., Xu, Y., 2020. Predominance of hexamethylated 6-methyl branched glycerol dialkyl glycerol tetraethers in the Mariana Trench: source and environmental implication. Biogeosci. 17, 2135-2148.

---

## Author Response (AR1)

**DEPARTMENT OF GEOSCIENCES**

*Programs in Geology, Geography,*
*Earth Systems*
**Morrill Science Center**
**627 North Pleasant Street**
**Amherst, MA 01003-9297**
**Tel: 413-545-2296**
**Fax: 413-545-1200**

November 20, 2021

Dear Dr. Zhengtang Guo,

We have now revised our manuscript "*Biomarker Proxy Records of Arctic Climate Change During the Mid-Pleistocene Transition from Lake El'gygytgyn (Far East Russia)*". In making our revisions we have tried to address all of the points the reviewers raised. Although there were a number of requested changes, in many cases these were relatively minor. Overall, the largest changes made to the manuscript include:

- o Adding additional requested information to the supplementary materials, including additional statistical analyses, and potting our data with additional lacustrine brGDGT calibrations.
- o Adding new discussion of the Walker Circulation in section 5.3, and its potential role in driving the half-precession cycle seen in our record, in light of a reference suggestion provided by one of the reviewers.
- o Expanding the discussion on other environmental factors that may impact brGDGT distributions.
- o Revising the discussion text to clarify our interpretation of changes in leaf wax distributions during the MPT.
- o Revising figure elements to meet all the guidelines of *Climate of the Past* formatting.

Please see the following pages for a point-to-point response to the reviewer suggestions. We hope that you now find our revised manuscript suitable for publication in Climate of the Past.

Sincerely,

Kurt R. Lindberg, on behalf of the co-authors

CoP Paper Comments and Responses

**Reviewer #1**

**General Comments**

*The MPT is a perplexing component of Quaternary climate change which provides a unique opportunity to understand the interconnectivity and feedbacks between different components of the climate system on orbital and sub-orbital timescales. Lindberg et al. present an interesting dataset from a unique archive to assess Arctic response to MPT climate dynamics and provide thorough discussion on the role of the North Pacific, Bering Sea, and inter-hemispheric teleconnections which could result in the observed changes in vegetation and temperature in north-eastern Russia. They properly outline the limitations of their datasets and calibrations, interpreting their data to a suitable resolution and using statistically robust techniques. The study concludes that the warm interglacial conditions widely observed during MIS 31, as well as subsequent MPT cooling, are not apparent at Lake El'gygytgyn. They discuss potential causes for this, as well as the detected sub-orbital cyclicity in the temperature record, including the interplay between temperature and moisture availability resulting from teleconnection with Arctic and sub-Arctic, as well as tropical and Atlantic Ocean feedbacks. Overall, I recommend that this manuscript be accepted subject to minor revisions. I have outlined some areas which I feel could benefit from additional discussion, as well as some technical corrections. I hope that the authors find these recommendations helpful.*

We thank Dr. Worne for providing a careful review of our manuscript and for the useful suggestions.

**Specific Comments**

*Could a short discussion be made on the difference between the results from the original and re-analysed samples from de Wet et al. (2016)?*

As the samples in this part of the core have low amounts of 6-methyl isomers, re-analyzing the samples with the newer HPLC methods yielded results similar to the original ones. Indeed, there is a very strong correlation between the original de Wet MBT values and the MBT values of the re-analyzed samples, with a slope near 1 (see new figure S7).

That said, the study of de Wet et al. (2016) applied an older MBT/CBT calibration from Sun et al. (2011). Since then, improved HPLC separation methods have revealed that the apparent dependence of brGDGT cyclization (the CBT index) on temperature was an artefact of the incomplete separation of brGDGT isomers. Furthermore, the Sun et al. (2011) calibration was also based on a substantially smaller set of lakes than the new BayMBT calibration that we apply here. While there is a good correlation between the reanalyzed BayMBT-based temperature values and the original MBT/CBT-based values, the slope of these data is approximately 0.5, indicating that that amplitude of temperature variability based on the BayMBT calibration is approximately half that of the MBT/CBT-based calibration previously reported. Given that brGDGTs are likely produced in-situ in Lake El'gygytgyn, and that today lake water exhibits a small temperature range (0 to 5-6∘C for shallow parts of the lake in summer), a calibration exhibiting a smaller overall amplitude is more realistic.

We added a supplemental figure showing how the HPLC methodologies compare (Fig. S7), and include text in both the main manuscript (Lines 217-220) and the supplement (section 1.2) discussing this comparison.

*Figure 5 presents an MBT/CBT calibration from Sun et al. (2011) in light brown squares, however there appears to be no mention of this calibration is made in the text. There is clearly an offset in values in the MBT/CBT record compared to the African lakes and BAYMBT calibrations at ~1.1 ma. Could the author include some information on this calibration and the likely reason for this discrepancy, as they have for the Greenland lakes calibration? This is particularly relevant for the later discussion on MIS 31 as the MBT/CBT calibration has notably higher temperatures which would support superinterglacial warmth, compared to the BAYMBT and African Lakes which do not show a similar peak in temperature or subsequent cooling trend.*

The MBT/CBT calibration of Sun et al. (2011) is included in this figure because is the calibration that de Wet et al. (2016) published their brGDGT data on. As this calibration is based on the older HPLC methods that did not separate the 5-methyl and 6-methyl isomers, it is not the best one to apply currently (we note that the de Wet et al. (2016) samples were mainly analyzed in 2014, prior to the newer HPLC method being published).  Indeed, based on the BayMBT temperature reconstruction, MIS 31 does not stand out as such a strong interglacial as it does in the MBT/CBT-based reconstruction. That said, if we apply the MBT/CBT-based reconstructions from the entire new-method record, we likewise see that MIS 31 would not stand out as a particularly warm interglacial (we do not show this, as there is no strong case for using MBT/CBT with the new HPLC method). We have added some information to the text about this calibration (Lines 320-332) and its offset from the others.

*The authors include a very interesting discussion on suborbital cyclicity at ~11 kyr. Some mention is given to the potential control of the monsoon. I wonder if the authors has considered the amplification of the Walker Circulation as a mechanism of suborbital cyclicity? Intensification of the Walker Circulation is suggested to have occurred in the build up to the MPT from ~1.17 Ma, propagating to the high latitudes through changes in the El Nino Southern Oscillation and East Asian Winter Monsoon (McClymont & Rosell-Melé, 2005; Stroynowski et al., 2017). Evidence from the adjacent Bering Sea supports this, where sea ice is suggested to have responded to the resultant changes in wind strength, temperature and moisture delivery (Stroynowski et al., 2017; Worne et al., 2021). This would also fit with the authors discussion of changing wind strength and wetter conditions.*

Thank you for this suggestion, we had not seen the Worne et al. (2021) paper yet. We agree that the Walker Circulation may contribute to some of the signals seen in the Lake El'gygytgyn brGDGT record. We have added a couple paragraphs of new discussion on how Walker Circulation may have contributed to the half-precession signal seen at Lake El'gygytgyn and influenced North Pacific climate in Section 5.3 (Lines 423-451).

*Line 437: Evidence from the Site U1343 does not show reduced diatom productivity, where the opal MAR record (Kim et al., 2014) is high through this interglacial, indicating high productivity. Furthermore, Detlef et al. (2018) states that "beginning at MIS 25, [Site U1343] is characterised by an ice-free eastern Bering Sea". Recent diatom data may be in better support of your*

*discussion here, where fossil assemblages suggests that MIS 25 represents an interval of peak marginal sea ice conditions across the MPT interval, suggested to be a result of increased wind strength and longer sea ice melt seasons (Worne et al., 2021).*

The climatic and oceanic changes that took place during Marine Isotope Stage 25 are complex, and we thank the reviewer for pointing out the need to improve the discussion around Site U1343 at that time. Indeed, high opal accumulation rates at Site U1343, together with the low sea-ice biomarkers may indicate highly productive, ice-free conditions. On the other hand, as the reviewer points out, recent results of Worne et al. (2021), based on diatom assemblages at Site U1343, suggest an increase in marginal sea-ice conditions, possibly related to a lengthened sea-ice melt season. We have revised this paragraph to better characterize the changes going on in the N. Pacific and how they may relate to temperature change at Lake El'gygytgyn (Lines 506-522).

**Technical Corrections**

*• Line 12: comma after "Arctic"*

Comma was added. (Abstract, Line 12)

*• Line 35: comma after "(Brigham-Grette et al., 2013)"*

Comma was added. (Abstract, Line 36)

*• Line 36: rephrase, perhaps to "tundra vegetation, where the tree line lies…"*

Text here has been revised. (Section 1, Line 37)

*• Line 67 and 75: capitalise Arctic*

Arctic is now capitalized. (Section 1, Lines 68 & 78)

*• Line 114 and 116: extra space before methanol needs removing.*

The extra space is removed. (Section 3.1.1, Line 125 & 127)

*• Section 3.1: be consistent with capitalisation of Eq or eq.*

CoP instructions specify that "Eq." should be used and this is changed accordingly. (Section 3.1.2, Lines 115-182)

*• Figure 3: Caption for E and F appears to have errors with references in the wrong place and text missing, perhaps because of referencing software. Needs to be re-written.*

The figure 3 caption has been revised/re-formatted to fix these technical errors.

*• Line 334: NE has been fully written as northeast earlier in the text, needs to be consistent.*

"northeast" is written out (Section 5.3, Lines 352-454).

• *Line 450: "short-lived" needs hyphenating.*

Hyphen is added. (Section 5.3.2, Line 528)

References cited here, not included in original manuscript:

Kim, S., Takahashi, K., Khim, B. K., Kanematsu, Y., Asahi, H., & Ravelo, A. C. (2014). Biogenic opal production changes during the Mid-Pleistocene Transition in the Bering Sea (IODP Expedition 323 Site U1343). Quaternary Research, 81(1), 151–157. https://doi.org/10.1016/j.yqres.2013.10.001

McClymont, E. L., & Rosell-Melé, A. (2005). Links between the onset of modern Walker circulation and the mid-Pleistocene climate transition. Geology, 33(5), 389–392. https://doi.org/10.1130/G21292.1

Stroynowski, Z., Abrantes, F., & Bruno, E. (2017). The response of the Bering Sea Gateway during the Mid-Pleistocene Transition. Palaeogeography, Palaeoclimatology, Palaeoecology, 485(March), 974–985. https://doi.org/10.1016/j.palaeo.2017.08.023

Worne, S., Stroynowski, Z., Kender, S., & Swann, G. E. A. (2021). Sea-ice response to climate change in the Bering Sea during the Mid-Pleistocene Transition. Quaternary Science Reviews, 259, 106918. https://doi.org/10.1016/j.quascirev.2021.106918

Citation: https://doi.org/10.5194/cp-2021-66-RC1

The references of Kim et al. (2014), McClymont and Rosell-Mele (2005), Worne et al., (2021) have now been added to the manuscript at appropriate places.

**Reviewer #2**

*Lindberg and co-authors report on the well-known and unique Lake E sediment record from the Russian Arctic. Their analysis of brGDGT and n-alkane biomarkers allow new and revised interpretations of past environmental change in terms of precipitation and vegetation across the mid-Pleistocene transition (MPT), a global and enigmatic period of climate change. Overall, the paper is very well written with clear accompanying figures that allow the reader to visualize the text appropriately. Although the authors do explore the utility of the brGDGT proxy using various calibrations and statistical analyses, I offer some suggestions for further data interrogation that I hope will benefit the paper. The fact that this paper provides a rare and continuous terrestrial Arctic paleoclimate record across the MPT naturally amplifies its impact. Collectively, with some revisions, this manuscript will be a highly valuable contribution to both the paleoclimate and brGDGT proxy community. I congratulate the authors on a very nice manuscript – it was a pleasure to read.*

We thank the reviewer for the positive comments and for providing a careful review of our manuscript.

**Specific Comments**

*L11: Please be consistent with ka or kyr throughout the paper. Same applies for Ma and Myr.*

Following the recommendations in "*Terminology of geological time: Establishment of a community standard*" by Aubrey et al. 2009 (Stratigraphy, vol. 6, no. 2), we use "ka" and "Ma" to represent geohistorical dates while "kyr" and "Myr" and used to represent geohistorical duration. We have corrected a couple of errors throughout the manuscript.

*L34-35: Maybe I'm just not familiar with ecology well enough, but it sounds oxymoronic to have cool forests indicative of a warm climate?*

Good point - we understand why this might be confusing. Presently, Lake El'gygytgyn is surrounded by tundra with the nearest trees ~150 km to the south. Thus, having any trees around the lake, even those representing cool forest species, indicates a significantly warmer climate. We re-worded this sentence for clarity (Section 1, Line 37).

*L40: Might be worth briefly mentioning what proxies these T and P reconstructions are based on.*

The previous temperature and precipitation reconstructions, which only represent short segments of the overall Lake El'gygytgyn record, are based on pollen assemblages. We added this information at line 41.

*L72: You say that MPT lake records are particularly rare. Are there any others? If so, would be worth adding some references, otherwise clarify that Lake E is the only known one!*

There are only a handful of lacustrine records continuously spanning the MPT, including Lake Baikal, Lake Malawi and Lake Ohrid, as only a relatively small number of lakes have been drilled to date. We revised this sentence to include appropriate citations and clarify that El'gygytgyn is the only such record from the Arctic (Section 1, Lines 73-75).

*L84: Can you provide a sentence or two on how the age model is constructed (i.e., what geochronological tools/proxies)? Do you have a sense of how the age model uncertainty (good/bad) effects the timing on your biomarker records? This seems particularly relevant for the spectral analyses.*

We utilize the age model of Nowaczyk et al. (2013), which is based on iterative tie-point identification using 1) paleomagnetic reversals, 2) comparison of biogenic silica to the LR04 oxygen isotope stack, and lastly 3) comparison of TOC and magnetic susceptibility to summer insolation. Given the resolution and uncertainty of the target curves (LR04 and insolation), the absolute age uncertainty could be off by 3-15 kyr, although the relative age assignments have a precision of 500 years (Nowaczyk et al. 2013). Since the Lake E age model is tuned to LR04, it may be circular to discuss leads and lags between datasets, and this is generally avoided in our discussion. Given the 3rd-order tie points improve the relative precision to ~500 years, it remains useful to discuss the spectral results because the multitude of Lake El'gygytgyn proxies appear to exhibit different orbital-scale patterns. We added a brief summary of these main points to the text about the age model (Section 2, Lines 87-94)

*L95: Please clarify which months you include in summer...JJA?*

Yes, we are referring to JJA for summer temperatures. We now specify this in the text (Section 2, Line 105).

*L96: When do shallower regions reach 5-6 degC...summer? Please specify.*

Shallower regions within Lake El'gygytgyn reach 5-6 degrees C during the summer. We now specify this in the text (Section 2, Line 107)

*L100: Thermodynamics of the lake? Please specify.*

In this sentence, we are referring to atmospheric temperatures at Lake El'gygytgyn. Nolan et al. (2013) analyzed how atmospheric pressure patterns may or may not relate to the temperature at Lake El'gygytgyn. They found that recent warming signal at El'gygytgyn was not attributed to changes in the frequency of different weather patterns, implying that general warming of the atmosphere (rather than storm tracks) controls air temperature. We revised this sentence for clarity (Section 2, Line 108-111).

*L106: It seems like these 41 re-analyzed samples are a sub-set of the original (de Wet et al., 2016). Was there a strategy for why these ones were chosen rather than re-analyzing the entire dataset?*

Yes, the re-analyzed samples are a subset of the original 174 samples that de Wet at al. (2016) analyzed. A complete re-analysis of the de Wet samples is not needed, as the higher-resolution sampling is not necessary for the orbital-scale questions investigated here. We did not re-run the full dataset due to the cost. Furthermore, as these samples do not contain significant amounts of the 6-methyl isomers, re-analyzing yields similar results (also see comments in response to Reviewer #1). Approximately every 3rd sample was re-analyzed. We added text clarifying how we selected samples for reanalysis (Section 3, Line 117). Furthermore, we added some discussion on how the reanalyzed samples compare with the original results (Supplemental section 1.3 and Fig. S7).

*L109: This sentence is a little misleading because I do not believe de Wet et al. (2016) analyzed n-alkanes. Please rephrase. In addition, and similar to my prior comment, these do not represent all samples de Wet et al. (2016) analyzed. Was there reason behind how many and which samples were re-analyzed for brGDGTs and n-alkanes from the original sample set of de Wet et al. (2016)?*

The doctoral thesis of de Wet (2017) does include *n*-alkane analyses. This is cited correctly. The de Wet et al. (2016) EPSL publication only includes GDGT data. de wet (2017) ran 85 samples for *n*-alkanes, which we supplement with the 127 samples that we newly extracted for this study. We made one small edit to try and make the distinction between de Wet's studies clear (Section 3, Line 120).

*L141: I do not agree with the decision to only use 5-methyl indices. Micro/mesocosm experiments for lake brGDGTs (Martínez-Sosa and Tierney, 2019) in addition to the environmental samples from East Africa (Russell et al., 2018) demonstrate a positive correlation between the abundance of total 6-methyl brGDGT isomers and temperature. A lake record from Iceland also shows some 6-methyl isomers strongly correlated with T inferred from alkenones (Harning et al., 2020). Since we do not have culture experiments to better test which brGDGTs are produced across which environmental gradients, I think it is best to explore the full gamut rather than a subset.*

In this portion of the El'gygytgyn record, the 6-methyl isomers average 9% of the total brGDGTs. However, we note that many Lake El'gygytgyn samples do not contain any 6-methyl isomers at all, particularly in the Late Pleistocene. Thus, when analyzing the entire 3.6 Ma record (which we are doing in other publications), we need to use the 5-methyl indices, and so focus on those indices. As the MBT'$_{5ME}$ yields glacial-interglacial cycles that are present in other Lake El'gygytgyn proxies and we can generate a continuous record using the 5-methyl brGDGTs, we stand by our decision to use MBT'$_{5ME}$.

*L145: Like my previous comment, I suggest, especially since we do not yet have culture studies*

*for brGDGTs and that there are not that many existing empirical lake calibrations that separate the 5 and 6-methyl isomers, all calibrations be tested. These would include:*
*Feng et al. (2019)*
*Harning et al. (2020)*
*Raberg et al. (2021)*
*Since they are all also from or include high latitude/altitude locations, they should be just as applicable as the East African and Greenland calibrations and may offer some interesting insights*
*(see later comments).*

We added the calibrations of Raberg et al. (2021) and Feng et al. (2019) to the manuscript (Lines 164-178, Eq. 5-7) and Supplemental Figure S8. However, Harning et al. (2020) developed a cross calibration between the alkenone-based UK37 Index and brGDGTs, which we cannot apply here as Lake El'gygytgyn does not contain alkenones.

*L158: Is 211 the number of samples you analyzed (127) and those from de Wet (85)? If so, should this number read 212?*

Yes, thank you for catching that. We corrected this (Section 4.1, Line 185)

*L164: CPI should be defined in the methods.*

The CPI equation is defined in the supplemental text. The text in section 4.1 (Line 191) now points the reader to the supplemental equation (Eq. S1).

*L166: Would it be possible to conduct a change point type analysis here? Otherwise, it seems rather arbitrary to select 900 ka as the boundary. I agree it looks like 900 ka reflects a regime shift, but our eyes can do weird things and I find that statistics help us be more objective.*

We conducted a change point analysis on our *n*-alkane data following the ramp-fitting methodology published by Capron et al. (2021, Nature Geoscience). Using this method, we found the significant ACL shift occurred between 960 and 930ka (95% CI: 1019-895 ka). When comparing with marine records in the North Pacific and North Atlantic, we retain the use of 900 ka as a general partition based on the end of the transition interval in the ACL record. However, 900 ka is still only an estimated average for the timing, globally, therefore we believe that our analysis of ACL before and after 900 ka to still be valid. We added a description of the change-point analysis and its results into the supplemental section (Supplement, Lines 19-22 and Figure S2), and altered the main text appropriately to refer the reader to that section (Lines 193-196).

*L224: Other Arctic sites not mentioned suggest this assumption may be less robust (e.g., Dion-Kirschner et al., 2020), where the data show that terrestrial plants also produce a substantial amount of mid-chain plant waxes. I also skimmed the Wilkie paper, and it looks like they only reported on alkanoic acids, which do not necessarily correlate with alkanes as implied here. Might be worth briefly expanding on some of these limitations for your ACL interpretation.*

We feel that our interpretation of ACL (Lines 258-290) as generally reflecting large-scale vegetation changes in the lake catchment is robust as explained in the manuscript text. Furthermore, this explanation is not substantially different from prior interpretations of ACL; as vegetation is largely sensitive to temperature and aridity, it follows that ACL would track vegetation changes and be related to both temperature and precipitation. We also note that we analyzed both *n*-alkanoic acids and *n*-alkanes on Lake El'gygytgyn samples (Habicht 2019, dissertation) and find similar results for both downcore concentrations and isotopic values for both compound classes (this work is in preparation for publication). The $C_{27}$ *n*-alkane and the $C_{26}$ *n*-alkanoic acid are the most abundant homologues.

Habicht, Mary Helen, "Middle to Late Pleistocene Paleoenvironmental Reconstructions from Lake El'gygytgyn, Arctic Russia" (2019). *Doctoral Dissertations*. https://doi.org/10.7275/15018735

*L229: How do these previous studies link ACL and aridity? Modern instrumental calibrations or downcore proxy correlations? I'm not as familiar with using alkanes as a precipitation proxy, so it may help other readers in a similar boat. Is there a physiological mechanism that causes this response to precipitation variability?*

Prior studies that have observed relationships between ACL and temperature or aridity have used other independent proxies (e.g., pollen, TEX86, leaf wax isotopes, lignin phenols) to determine temperature or aridity changes. This is now noted in the main manuscript text at Lines 265-269.

*L233-234: Same comment as above. What proxies is this aridity condition based on? Please briefly mention.*

We revised the text to reflect how these other studies have been able to compare the ACL changes to independent aridity metrics, namely leaf wax isotopes or pollen (Section 5.1, Line 265-273).

*L249: In re to the visual correlation between ACL and landscape openness index, are there any statistical regression techniques that could be used to support your observation?*

We have now performed regression analysis between ACL and landscape openness, confirming our qualitative description of how these two variables are negatively correlated. We recognize that a visual presentation of the correlation would be useful, and added this regression (Figure S3) with appropriate text in the supplement, and edited the main text (Line 196) to refer the readers to this section.

*L251-252: Since you mention at the end of the previous paragraph that you refine the ACL interpretation for Lake E, does your interpretation differ from the prior studies you discussed? If so, in what way?*

While some prior studies have noted ACL tracks changes in temperature or aridity, we think most authors would agree that this is an oversimplification and multiple factors are responsible for driving vegetation change. We note that many prior studies do not have both pollen and leaf wax data. In this study, the addition of pollen data shows us that ACL broadly tracks the landscape openness index based on pollen assemblage data. Our interpretation of ACL does not differ substantially from that of prior studies. For example, Keisling et al. (2017) previously reported that aridity, temperature and vegetation change drive changes in ACL. How our interpretation compares to other studies is detailed in the paragraphs beginning at lines 258-290.

*L257-259: I don't agree with the argument that dominance of 5-methyls suggests MBT'5Me is most suitable. Transfer functions for other non-biomarker proxies (e.g., foraminifera, chironomids, etc.) can include non-dominant taxa that are important for interpreting changes in T, or whatever the variable of interest is. Therefore, I'd suggest exploring all the available calibrations that separate the isomers, even those that may include 6-methyls, and those that include other calibration approaches, such as stepwise forward selection, as I suggest earlier. I'll also note that Russell et al. (2018) present a SFS calibration that does feature a lower RMSE than their MBT'5Me index calibration's.*

As noted above in the response to line 141, we cannot apply the 6-methyl brGDGT indices to many samples in this part of the Lake El'gygytgyn record as only 5-methyl brGDGTs are present. In order to examine the full record (we are exploring this in other publications) and generate a continuous temperature reconstruction, we need to use 5-methyl brGDGTs. Furthermore, we have explored indices based on 6-methyl brGDGTs and they do not yield any meaningful information as far as we can tell (e.g., glacial-interglacial cycles are not revealed). We note that the SFS calibration of Russell et al. (2018) includes brGDGT IIb' but as this brGDGT is typically not present in Lake El'gygytgyn samples, we did not apply this calibration.

*L261: Fair to exclude Dang et al. (2018). And if the other calibrations I suggest are also not optimal is some similar way, it could be mentioned here, but each should be systematically evaluated.*

In section 5.2 (Lines 292-341), we expanded this section slightly to include the calibrations of Raberg et al. (2021) and Feng et al. (2019). We also expanded the results and supplemental discussion to further discuss how the various calibrations compare, now including the Raberg et al. (2021) and the Feng et al. (2019) calibrations (see Figure S8).

*L265: Raberg et al. (2021) provide a number of Arctic sites in their modern lake brGDGT calibration.*

In this sentence, we only meant that the BayMBT training set does not include many Arctic sites outside of Alaska, not that such datasets don't exist. We edited this for clarity. It may be beneficial to incorporate the data from Raberg (2021) into the BayMBT training set, but we did not think it necessary for this study, as the BayMBT already contains lakes from a wide variety of

environments.  Also, this is beyond the scope of our study, which is not a calibration study but a paleoclimate reconstruction.

*L270: Yes, naturally because they are all use the same index. I think this is also one reason why it will be interesting to try some other non-MBT-index calibrations to test if these patterns are consistent or not and explore what brGDGTs are key drivers in the indices. In this sense, it may also be interesting to conduct a Pearson correlation matrix as Feng et al. (2019) did to explore the relationship between individual brGDGTs. You may find similarities between other calibration Pearson matrices that could support use (or not) of a certain calibration.*

We generated a brGDGT correlation matrix and compared it with similar matrices from other datasets. Broadly, we find from the Lake El'gygytgyn dataset resembles the BayMBT calibration dataset, further supporting our use of the BayMBT calibration.

*L276-293: This paragraph seems to imply that the pollen-inferred T record is more reliable. Both pollen and brGDGT T proxies have various assumptions and therefore only reflect approximations of past events. I think this paragraph could be rephrased to exclude statements such as "unreasonably large, L280" and "more realistic, L284" and perhaps more objectively compare the 2 T proxy records (pollen and brGDGTs). One of the motivations for this study seems to be producing a rare continuous T record through the MPT. However, if we already have that through pollen, what's the value of brGDGTs here, especially if you think that are not realistic? There are many brGDGT studies that link different distributions to various environmental parameters (e.g., pH, salinity, DO, etc.), which may be a more valuable, or at least supplemental, discussion topic to include.*

We have revised the wording of some of these sentences (now at lines 326-330). We think it is an interesting point that Lake El'gygytgyn lake temperatures do not change much over seasonal cycles (Nolan & Brigham-Grette, 2007), which might have a buffering effect on glacial-interglacial summer temperature variability in the brGDGTs as well. Because of this, we retain some of the wording comparing the brGDGT temperature calibrations investigated in this study.

We note that that pollen data only exists for certain portions of the Lake El'gygytgyn record; for much of the Lake El'gygytgyn core, pollen analyses are still in progress. One advantage of brGDGTs, is that the method is faster compared to pollen counting. We note that the pollen data spanning the MPT provides a landscape openness index but the authors did not provide a pollen-based temperature reconstruction (although pollen-based temperature estimates are available for certain discrete parts of the record, such as MIS 5 and MIS 11 (Melles et al., 2012)). Therefore, brGDGTs provide the first continuous temperature reconstruction from Lake El'gygytgyn spanning the MPT. While, for the reasons described in the text, we caution readers against interpreting the absolute temperature values yielded by brGDGTs, we believe the warming and cooling trends in these data are robust and note that other proxies from Lake El'gygytgyn support our brGDGT temperature record. Certainly, other environmental factors are likely influencing the brGDGT record and we note the limitations of brGDGTs in many places throughout the manuscript (e.g., one place where this is discussed is at lines 473-492).

*L286: In Figure S5, it would be interesting to plot up some additional environments as well. Just eyeballing, there appear to be some similarities between brGDGT distributions in Lake E and Svalbard fjord sediments (Dearing Crampton-Flood et al., 2019) and marine sediments (see plots in Xiao et al., 2020), as examples.*

As Lake El'gygytgyn is a freshwater lake and has been a freshwater lake throughout its entire history, we feel it is appropriate to limit our comparison to other lacustrine datasets, and so do not incorporate the marine brGDGT datasets into the ternary diagram shown in Figure S5.

*L318: Do other qualitative or quantitative climate records from Lake E show a long-term cooling trend that could be compared here?*

One of the novel aspects of our study is that it is one of the first continuous temperature reconstructions from Lake El'gygytgyn. There are numerous other proxies from this core spanning the MPT. While none show a clear cooling during the MPT, we also note that none are direct temperature proxies. This information is now stated at lines 369-373.

*L321-322: Yes, a lack of MPT cooling at Lake E may be odd, but in addition to climate-driven hypotheses, our still growing knowledge of the brGDGT proxy may also limit or obscure these observations.*

We agree and feel that we have clearly noted the limitations of brGDGT data throughout the manuscript.

*L413: Or there is another environmental or microbial factor that contributes to brGDGT distribution changes (e.g., Weber et al., 2018; De Jonge et al., 2019, 2021) and Facies C is still indicative of T.*

It is certainly possible that other factors contribute to brGDGT distribution changes. Indeed, changes in redox might be expected to impact the brGDGT temperature reconstruction. Yet, available information would suggest that the more highly-oxidized environment of MIS 31 could introduce a warming bias in those temperatures (e.g., Buckles et al. 2014, Martínez-Sosa and Tierney 2019). This surprisingly contrasts with what we see, in that MIS 31 does not look particularly warm. We added some text addressing this specific issue at lines 473-492.

While other environmental factors may contribute to brGDGT distribution changes, the advantage of the BayMBT calibration is that it spans a wide range of environmental conditions, and so these other potential factors are ideally contained within the calibration uncertainty which we report.

*L440: You explain ACL earlier as a proxy for aridity, so might be better to limit your interpretations of ACL to that here as well. Why would trees expand under increased aridity?*

Based on the Lake El'gygytgyn pollen record published in Zhao et al. (2018), the amount of tree and shrub pollen increases during MIS 25, which is paired with a decrease in *n*-alkane ACL at this time. Based on our interpretation of ACL increasing with increasing aridity and cooler brGDGT-inferred temperatures, the observed ACL decrease during MIS 25 suggests wetter conditions in the region, supporting the expansion of trees and shrubs. We have revised this sentence to indicate that the decline in ACL coincides with an expansion of trees and shrubs in the pollen record (Lines 509-513).

*L492: Clarify that this is increased "oceanic" stratification. It took me a few reads to realize you weren't referring to Lake E.*

"oceanic" is inserted (Section 5.4, Line 571)

*L499: Can you specify which SST proxies? This makes it easier for the reader to independently compare different proxies, as each have different assumptions and interpretations.*

We specified in the text what proxies are being utilized at the various N. Pacific paleoceanographic sites (Lines 576-588). These include alkenones, TEX$_{86}$, diatom assemblages and opal mass accumulation rates, and nitrogen-isotope records of nutrient utilization. These are likewise included in the Y-axis labels and caption for Figure 6.

*Figure 1: Panel B shows 2 yellow dots but only one is mentioned in the caption. Please clarify.*

We revised the figure to only include the location of the drill core 5011-1 analyzed in this study. We removed the yellow dot showing the location of piston core LZ1024, which spans the past ~300 ka but is not discussed in this manuscript. Thank you for catching this.

*Figure 3: Please clarify in the caption what the bold line is for n-alkane ACL...also a 5-point moving*
*Average?*

We clarified this in the caption. Yes, it is a 5-point running mean. (Note that additional formatting changes were done on this and other figures to align the formatting with *Climate of the Past* guidelines.)

*Figure S5: Would be good to include all the lake sediment samples I mention from the other calibration studies in the ternary diagram.*

In the ternary diagram (Fig. S5), we added the brGDGT data from the calibration study of Raberg et al. (2021). Their full calibration dataset contains several other datasets that overlap with the BayMBT suite, and so we only add the new data from the Icelandic lakes.  Likewise, the Feng calibration is based on other datasets, mostly included within BayMBT, and so do not add those points to the figure. Harning et al. (2020) include a single lake surface sediment sample which

could be compared, but given its similarity to the other lakes data, we did not feel it necessary to include this datapoint in Figure S5.

**Technical Comments**
*L22: Change "exhibits" to "exhibit". Data are plural.*

"Exhibits" has been changed to "exhibit". (Abstract, Line 23)

*L46: Comma after "ka".*

Comma was added. (Line 46)

*L60: Insert "may have" or something similar before "worked". Otherwise, it sounds like this is what happened.*

The wording in this sentence has been adjusted. (Line 60)

*L75: Capitalize "arctic", and check if needed elsewhere throughout the ms.*

Arctic has been capitalized. (Line 78)

*L106: I suggest phrasing as "46.77 to 31.06 m" so that it is consistent with the ages you mention just before (i.e., older/deeper to younger/shallower).*

Ordering change has been made. (Line 116)

*L161: Change "they" to "the peaks" or something. It's always easier to follow if pronouns are not used.*

The wording in this sentence has been adjusted. (Line 188)

*L167: Extra and/or missing words around "samples characterized". Please clarify.*

The wording in this sentence has been adjusted. (Line 194)

*L362: Journal should not be referenced in in-text citations.*

The journal name has been removed from this reference. (Line 417)

*L389: Does it need to be mentioned or clarified that the LR stack is annual rather than summer biased as you argue the brGDGTs are?*

The LR04 stack reflects a 2,000-year climate average at each interval.

*L393-394: Are there any statistical regression analyses that can be used to support this observed*

*alignment between brGDGTs and obliquity?*

We argue that brGDGTs at Lake El'gygytgyn are aligned with obliquity because the spectral analysis of our record shows a strong 41 kyr signal (Fig. 4a) in line with the period of obliquity.

*L412: There are some missing words in the latter half of this sentence.*

The wording in this sentence has been adjusted. (Lines 481-482)

*L539: A little wordy/awkward phrasing…perhaps just "Spectral analysis of the brGDGT record, in comparison with marine records, suggests…"*

The wording in this sentence has been adjusted. (Line 619)

*L506: Add "Sea" after "Bering"*

"Sea" has been added after "Bering". (Line 585)

**References (Not included in original text)**

Dearing Crampton-Flood, E., Peterse, F., Sinninghe Damste, J.S., 2019. Production of branched tetraethers in the marine realm: Svalbard fjord sediments revisited. Org. Geochem. 138, 103907.

De Jonge, C., Radujkovic, R., Sigurdsson, B. D., Weedon, J. T., Janssens, I., Peterse, F., 2019. Lipid biomarker temperature proxy responds to abrupt shift in the bacterial community composition in geothermally heated soils. Org. Geochem., 137, 103897.

De Jonge, C., Kuramae, E.E., Radujkovic, D., Weedon, J.T., Janssens, I.A., Peterse, F., 2021. The influence of soil chemistry on branched tetraether lipids in mid- and high latitude soils: Implications for brGDGT-based paleothermometry. Geochim. Cosmochim. Ac. 310, 92-112.

Dion-Kirschner, H., McFarlin, J.M., Masterson, A.L., Axford, Y., Osburn, M.R., 2020. Modern constraints on the sources and climate signals recorded by sedimentary plant waxes in west Greenland. Geochim. Cosmochim. Ac. 286, 336-354.

Feng, X., Zhao, C., D'Andrea, W.J., Liang, J., Zhou, A., Shen, J., 2019. Temperature fluctuations during the Common Era in subtropical southwestern China inferred from brGDGTs in a remote alpine lake. Earth Planet. Sci. Lett. 510, 26–36.

Harning, D.J., Curtin, L., Geirsdóttir, Á., D'Andrea, W.J., Miller, G.H., and Sepúlveda, J., 2020. Lipid biomarkers quantify Holocene summer temperature and ice cap sensitivity in Icelandic lakes. Geophys. Res. Lett. 47, 1–11.

Martínez-Sosa, P., Tierney, J., 2019. Lacustrine brGDGT response to microcosm and mesocosm incubations. Org. Geochem. 127, 12–22.

Weber, Y., Sinninghe Damste, J.S., Zopfi, J., De Jonge, C., Gilli, A., Schubert, C.J., et al., 2018. Redox-dependent niche differentiation provides evidence for multiple bacterial sources of glycerol tetraether lipids in lakes. Proc. Natl Acad. Sci. 115, 10926–10931.

Xiao, W., Wang, Y., Liu, Y., Zhang, X., Shi, L., Xu, Y., 2020. Predominance of hexamethylated 6-

methyl branched glycerol dialkyl glycerol tetraethers in the Mariana Trench: source and environmental implication. Biogeosci. 17, 2135-2148.

We have added the references of Feng et al. (2019), Harning et al. (2020), and Martínez Sosa and Tierney (2019) to the manuscript. Some of these other references are not included as they refer to marine or soil GDGTs, which are not directly relevant to the manuscript, as multiple lines of evidence point to lacustrine brGDGT production.  To help clarify the lacustrine source, we did add a different citation not listed here, to Kusch et al. (2019, Org. Geochem.), who's results of Siberian soil brGDGT distributions clearly differ from our Lake El'gygytgyn data.

---

## Author Response (AR2)

**DEPARTMENT OF GEOSCIENCES**

*Programs in Geology, Geography,*
*Earth Systems*
**Morrill Science Center**
**627 North Pleasant Street**
**Amherst, MA 01003-9297**
**Tel: 413-545-2296**
**Fax: 413-545-1200**

February 02, 2022

Dear Dr. Zhengtang Guo,

Thank you for your efforts as editor of this manuscript *"Biomarker Proxy Records of Arctic Climate Change During the Mid-Pleistocene Transition from Lake El'gygytgyn (Far East Russia)".* We are pleased to make the minor changed noted by the reviewers, which are found in this revised version. A point-by-point response to those suggestions is also attached.

We hope that you now find our further revised manuscript suitable for publication in Climate of the Past, and we look forward to seeing this in press.

Sincerely,

Kurt R. Lindberg, on behalf of the co-authors

CoP Paper Comments and Responses

**Reviewer #1**

*The authors have appropriately responded to all comments, and I thank them for considering my suggestions in their revised manuscript. I particularly commend the additional discussion about the Walker circulation with respect to their dataset in a regional context. Overall I recommend that this manuscript be accepted, at the discretion of the editor, with the exception of a few very minor changes:*

We thank Dr. Worne for again providing a careful review of our revised manuscript and for the helpful suggestions.

**Suggested Corrections:**

*• Line 483: Addition or substitution of reference to Stroynowski et al., 2017 (http://dx.doi.org/10.1016/j.palaeo.2017.08.023).*

We have included the reference to Stroynowski et al., 2017 in addition our reference to Worne et al., 2020 in our discussion about upwelling in the Bering Sea. (Section 5, Line 583)

*• Line 517: "exhibits a cool MIS 25"? or rephrase.*

We have rephrased this to read "exhibits a relatively cool MIS 25". (Section 5, Line 517)

*• Line 514: capitalize "Site"?*

"Site" has now been capitalized in referencing Site U1343. (Section 5, Line 517)

**Reviewer #2**

Reviewer #2 has recommended that our revised manuscript be accepted as is and did not suggest additional revisions. We thank the reviewer for providing another careful review of our manuscript.